# Untapped Resources: 10- to 13-Year-Old Primary Schoolchildren’s Views on Additional Physical Activity in the School Setting: A Focus Group Study

**DOI:** 10.3390/ijerph15122713

**Published:** 2018-12-01

**Authors:** Vera van den Berg, Eline E. Vos, Renate H. M. de Groot, Amika S. Singh, Mai J. M. Chinapaw

**Affiliations:** 1Amsterdam UMC, Amsterdam Public Health, Department of Public and Occupational Health, Vrije Universiteit Amsterdam, de Boelelaan 1117, 1081 HV Amsterdam, The Netherlands; v.vandenberg@vumc.nl (V.v.d.B.); e.vos2@vumc.nl (E.E.V.); m.chinapaw@vumc.nl (M.J.M.C.); 2Welten Institute-Research Centre for Learning, Teaching and Technology, Open University of the Netherlands, 6419 AT Heerlen, The Netherlands; renate.degroot@ou.nl; 3Department of Complex Genetics, School for Nutrition, Toxicology and Metabolism/Faculty of Health, Medicine and Life Sciences, Maastricht University, 6200 MD Maastricht, The Netherlands

**Keywords:** physical activity, school, physical education, implementation, perceptions children, intervention development, feasibility, qualitative research, preadolescents

## Abstract

Schools are considered ideal venues to promote physical activity (PA) in children. However, a knowledge gap exists on how to adequately integrate PA into the school day and in particular, on the preferences of children regarding additional PA in school. Therefore, the aim of our qualitative study was to gain comprehensive insight into 10–13-year-old primary schoolchildren’s perspectives on how to increase PA in the school setting. We conducted nine focus groups (32 girls and 20 boys) with children attending the final two grades of primary school in the Netherlands. We used inductive thematic analysis to analyze the data. The results showed that children were enthusiastic about additional PA in school. Children suggested various ways to increase PA, including more time for PA in the existing curriculum, e.g., physical education (PE), recess, and occasional activities, such as field trips or sports days; school playground adaptation; improving the content of PE; and implementing short PA breaks and physically active academic lessons. Children emphasized variation and being given a voice in their PA participation as a prerequisite to keep PA enjoyable and interesting in the long term. Finally, children mentioned the role of the teacher and making efforts to accommodate all children and their different preferences as important. Children have concrete ideas, acknowledging the challenges that accompany integrating additional PA in school. We therefore recommend actively involving children in efforts to increase school-based PA and to make “additional PA in school” a shared project of teachers and students.

## 1. Introduction

Although the numerous physical and mental health benefits of physical activity (PA) for children are scientifically well-documented [1,2], many children worldwide continue to fall short of meeting the recommended minimum of 60 min of moderate-to-vigorous physical activity (MVPA) per day [3,4,5]. Self-reported PA data for children from 39 countries show that only 23% of 11-year-olds and 19% of 13-year-olds currently meet this guideline [6]. These low levels of PA become even more pressing since time spent on MVPA typically further declines from primary to secondary education [7,8]. Moreover, it has been shown that a decrease in PA goes hand in hand with a decrease in motivation to keep participating in PA for children in the late years of primary school [9,10].

Because of their potential to reach many children, schools have been widely recognized as an ideal setting to contribute to raising children’s PA levels [11]. To this end, schools have been the nexus of the implementation of a variety of curricular and extra-curricular PA interventions, ranging from spending more time on physical education (PE), school sports, and PA in recess, to integrating movement into classroom time (e.g., classroom-based activity breaks during or in between lessons or physically active academic lessons) [12]. 

However, several systematic reviews concluded that school-based PA interventions often have only limited or short-term effects on children’s overall PA levels [13,14]. A possible explanation for this finding is that the evaluated interventions are commonly developed with limited involvement of school staff and children and may therefore be insufficiently tailored to their specific circumstances, needs, and wishes [15,16,17]. This could result in the development of inappropriate interventions resulting in sub-optimal implementation of and participation in these interventions, limiting their effectiveness [16,18,19]. In order to ensure that school-based PA interventions will succeed in daily school practice, they need to be relevant to local school contexts and appropriate for and attractive to the intended users (i.e. children and school staff) [16,17,19,20,21]. A recent systematic review concluded that involving students in the development and implementation of school-based health activities has positive effects on students’ motivation and attitude towards the activities, health-related outcomes, and social interactions, as well as the school’s culture, climate, rules, and policies [22]. Moreover, an earlier systematic review by Jacquez et al. (2013) has shown that involving children and adolescents in research projects can lead to increased quality of research and higher chances of translating research outcomes into action in daily practice [17].

In the past few years, several qualitative studies have investigated the perspectives of teachers and school administrators regarding primary school-based PA interventions (e.g., References [23,24,25,26]). These studies provided valuable information on facilitators and barriers, mostly regarding the organization and execution of PA. However, children are the ones who are expected to actually participate in the intervention, and it is very likely that their perspectives differ from the views of adult school staff. 

Thus far, several qualitative studies investigating children’s preferences regarding PA have been conducted, focusing on PA promotion in general (i.e., not specifically in the school setting). These studies identified a range of promising PA intervention components that were important according to primary school-aged children, such as providing new and fun activities, the use of technology, using rewards and incentives to encourage participation in PA, and providing opportunities to be active with friends [27,28,29,30,31,32]. Furthermore, a review of qualitative studies found that children prefer to choose their own physical activities and that a range of activities should be offered based on their preferences [15]. In addition, to encourage PA in 13- and 14-year-olds, adolescents recommended providing them with opportunities to try new activities, using role models to deliver the intervention, and adding rewards and elements of competition to PA interventions [33]. Although some of the results of the above-mentioned studies could be applied to PA interventions in schools, more specific and comprehensive insight is needed regarding PA promotion in the school setting. Accordingly, Carlin et al. (2015) emphasized that there is a gap in knowledge on how to maximize the potential of the school day in making children more active [27]. 

There is currently a lack of qualitative studies that focus specifically on PA opportunities in the primary school setting, and in particular involving the critical age group of 10–13 years, where, as mentioned above, PA tends to decline [7,8]. Moreover, the qualitative studies investigating additional PA opportunities that have been conducted in the primary school setting often have a narrow focus, determined in advance by the researchers, such as (barriers to) PA promotion in recess (e.g., References [34,35,36]), or adapting the school physical environment (e.g., References [37,38,39]). For example, a focus group study of Hyndman (2016) investigated what kind of school PA facilities 10–13-year-old children find appealing (e.g., a variety of recreational, sporting, and adventure facilities) [37]. However, this study was limited to features of the physical environment and did not consider other ways of enhancing school-based PA. As previous studies focused on certain predefined aspects of the school day, it remains unclear how exactly PA opportunities can be increased within the broader primary school setting according to children. In particular, knowledge is lacking on what kind of activities would be appealing to children, as well as what they need to successfully engage in school-based PA. Therefore, a comprehensive investigation of children’s thoughts and ideas concerning PA promotion within the primary school setting is required.

Additionally, the sustainability of PA interventions remains a major challenge [40,41]. There is currently little known about how to keep children under the age of 13 motivated to being active in the longer run [42]. For example, it has been shown that a major prerequisite for children’s engagement in activities is “having fun” [15,16,43]. Yet, we know little about what kind of PA programs result in sustained enjoyment instead of just momentary fun [16]. In this respect, Agans et al. pose that it is key to create contexts that support “positive movement experiences” for all children [44]. Early positive experiences with PA for children (i.e., focusing purely on enjoyment rather than increasing performance, and with a good “fit” between the context of PA and an individual’s skill level and preferences) result in a higher likelihood of continued PA participation in later life [44]. Therefore, it is imperative to investigate what PA characteristics contribute to positive movement experiences, for whom, when, and under what circumstances. 

In sum, for the development of attractive and effective PA interventions in primary school, more knowledge of the perspectives, needs, and wishes of children is needed. To the best of our knowledge, no qualitative studies have explicitly and comprehensively explored the multi-faceted ideas of 10–13-year-old children regarding increasing PA in the primary school setting. Therefore, the aim of our study was to: (1) explore 10–13-year-old primary schoolchildren’s perspectives on incorporating additional PA into the school day, in particular what practical ideas children have to be more physically active in school and what they think is important to successfully implement these ideas; and (2) to investigate children’s views on how to keep the proposed activities and ideas interesting and fun for a longer period of time. The results of this study can inform both research and practice regarding the development and implementation of future PA interventions aimed at 10–13-year-old children.

## 2. Materials and Methods

### 2.1. Study Design

We conducted a qualitative study, employing focus group interviews with 10–13-year-old children in the last two years of primary school in the Netherlands, i.e., in grades 5, 6, or combined 5/6. Focus groups are an effective method to explore the ideas and perspectives of children [45,46]. By interviewing children in a group setting at school, we aimed to create a familiar and safe peer environment, encouraging the children to share their ideas and thoughts [45,47], acknowledging them as experts [48], and reducing the power imbalance that may exist between children and the (adult) researchers [49].

### 2.2. PA in the Dutch Primary School System 

In the Netherlands, children aged 4 to 12 attend primary school (2 years of pre-school followed by grades 1 to 6). On average, children participate in an hour and a half of PE per week. There are no legal provisions issued by the government on the amount of PE that should be provided, but the guideline of the Ministry of Education, Culture and Science prescribes at least two to three lessons of PE per week. PE should either be taught by classroom teachers with a qualification for PE, or a PE specialist [50]. Schools are free to choose their own recess hours, e.g., a short break in the morning and a longer one for lunch break, or two breaks of the same length [51]. Furthermore, schools receive a budget issued by the government that they can spend at their discretion on facilities or personnel, such as a PE teacher [52]. In addition to the regular PA curriculum, there are some national/municipal initiatives (e.g., Healthy Schools, Jump-In, PLAYgrounds, Young People on Healthy Weight (JOGG)) that promote extra physical activity in schools, e.g., by providing PA interventions, school (playground) adaptations, or external coaches who organize PA activities during recess or after school hours.

### 2.3. Recruitment of Participants

We used a purposive sampling method [53] to recruit participants for our study. We approached 45 regular primary schools via email and follow-up phone calls, both from the network of the research group and schools that had not been approached before. Schools were located in both urban and rural areas across the Netherlands. To prevent children’s ideas being influenced by earlier implemented PA programs and/or initiatives that promote PAin school (e.g., Healthy Schools, Jump-In, PLAYgrounds, JOGG), only schools that did not structurally provide such additional PA programs were eligible to participate in our study. Nineteen schools did not respond (42%), and twenty-two schools (49%) declined to participate due to a lack of time (*n* = 9), an overload of research requests (*n* = 3), or without a specific reason (*n* = 10). Four schools (9%) agreed to participate.

After a school decided to participate, children in grades 5, 6, and 5/6, and their parents and/or caregivers received an information letter including information on the aim and procedures of the study, and an informed consent form. In order to include perspectives of children with different PA levels in each focus group, we asked parents to fill out a short questionnaire. The questionnaire contained three questions on their child’s PA behavior: (1) sports participation: yes/no; (2) hours of sports participation per week; and (3) hours spent in other PA per week, such as outdoor play and active gaming. We used this information to compose the focus groups (see below). For participation in the focus groups, at least one parent/caregiver and children older than twelve had to sign informed consent. Children received a small present (e.g., a key cord) for their participation afterwards. The Medical Ethical Committee of the VU University Medical Center Amsterdam approved the study protocol (2014.363).

### 2.4. Focus Group Composition

Group sizes of four to eight children allow for a manageable, active discussion, and have been recommended in focus group research with children [54,55]. In addition, single-sex focus groups are often recommended as boys and girls may have different attitudes, interests, and viewpoints [47,48], which could cause participants to feel uncomfortable to express their ideas in the presence of the opposite sex [47]. Therefore, we selected six boys and six girls in each class to participate in separate focus groups. All children that returned signed informed consent forms were eligible to participate. To include the perspectives of children with different PA levels, we randomly selected two children that scored relatively low, moderate, and high on PA behavior, as reported by their parents in the questionnaire (see above). In some classes only four to eight children returned the informed consent form, in which case they were all selected for participation. 

### 2.5. Focus Group Guide

The focus group interview guide can be found in Table 1. It was developed following the suggestions of Krueger and Casey [56], and consisted of semi-structured, open-ended questions combined with a task-based activity. To introduce the subject and to establish a relationship between the children and researchers, we started the focus group with two warm-up questions. To aid the generation of PA ideas, for main question two in the interview guide we applied a task-based activity in which we asked children to write down each idea on an individual sticky note (see Table 1). Afterwards, the children clustered their ideas in groups on a poster and discussed them with the group. By using this method we assured that all children had time to think about their answers without being influenced by others, and that all children had equal opportunities to express their ideas [57]. Probes and follow-up questions were used throughout the focus group to facilitate the discussion [58]. We ended the focus group with a summary and asked the children if the poster with their ideas was complete or if they had anything more to add. 

We developed a protocol and pilot-tested the focus group guide with a group of six boys and a group of six girls from a school that was not included in this study. Based on the feedback of the children and our own experiences, we adapted some instructions and the order of the two warm-up questions. The pilot confirmed that all questions could be answered within 45 to 60 min, which is an optimal duration when conducting focus groups with 10–14-year-olds [45].

### 2.6. Data Collection 

Five focus groups with boys and five focus groups with girls were conducted by two interviewers, i.e., Vera van den Berg (V.B.) and a research assistant. Both completed several courses on interviewing techniques during their educational programs, were trained in using the focus group guide, and practiced their moderator skills during the pilot focus groups. V.B. is experienced in working with groups of children and supervised the research assistant during the data collection. The researcher and the research assistant acted alternately as the group moderator who facilitated and encouraged the discussion, and as the observer taking notes. 

The focus groups were conducted at the schools of the children in a quiet and private room. For descriptive purposes, we asked the children to report their date of birth. We explained the procedures of the focus group, such as the duration, the use of audio equipment, and confidentiality/privacy. All children were encouraged to express their ideas and opinions. We emphasized that there were no wrong answers and that they were the experts on their preferences with regard to PA in the school setting.

The focus groups were audio-recorded and transcribed verbatim by V.B. and the research assistant. The average duration of the focus group was 46 min (range 35 to 50 min) for the boys, and 50 min (range 36 to 65 min) for girls. Raw data consisted of 312 pages of transcript (Calibri, 11, single spaced). 

### 2.7. Data Analysis 

The data analysis started after data collection was finished. The data was coded and analyzed using the qualitative data analysis software program ATLAS.ti version 7 (ATLAS.ti GmbH, Berlin, Germany). For the data analysis we followed the six steps of inductive thematic analysis of Braun and Clarke (2006) [59], and we used the checklist for qualitative reporting of interviews and focus groups as proposed by Tong et al. [60] to report the data. Thematic analysis as proposed by Braun and Clarke is a widely-used method for qualitative analysis for systematically identifying, analyzing, and reporting patterns (themes) within data (see audit trail in Table 2). To make sure children were unhindered by researchers’ previous notions about what might or might not be important in PA interventions, in our study we chose a “bottom-up” inductive approach without an a priori theoretical framework to guide and/or influence children’s answers [61]. As thematic analysis is not theoretically bounded, it is a suitable choice of method when using an inductive approach. Based on the recommendations of Elo and colleagues [58], one researcher (Eline E. Vos (E.V.)) coded the transcripts, and a second researcher (V.B.) checked all codes and complemented the files. E.V. and V.B. discussed all steps and outcomes in each phase of the data analysis process until they reached consensus. In case of disagreements and uncertainties, a senior member (Amika S. Singh (A.S.) or Mai J. M. Chinapaw (M.C.)) was consulted. We analyzed the data of boys and girls separately until data saturation was reached, i.e., that no new ideas were expressed for each of the groups, indicating that analyzing additional data would not substantially alter the initial categories that were generated [62].

## 3. Results

### 3.1. Participant Characteristics

Four primary schools agreed to participate; two located in urban areas and two in rural areas. Data of 52 children (32 girls from five focus groups and 20 boys from four focus groups) were included in the data analysis. Three focus groups were held with children in grade 5, two with children in a combined grade 5/6, and four with children in grade 6. Five of the focus groups consisted of six girls or boys, one of eight girls, one of four boys, and two of five boys. The mean age of the children was 11.7 years (SD = 0.6), ranging from 10 to 13 years. Eleven children were considered relatively high-active by their parents, 16 children medium-active, and 16 children relatively low-active. Data on activity-levels of the other 9 children were not provided. Overall, the data analysis revealed that themes were quite similar for girls and boys. Therefore, we present the data as a whole and highlight some cases in which the opinions of girls and boys differed.

### 3.2. Focus Group Results

We identified six key themes: (1) children’s motivations for participating in additional PA, (2) children’s ideas and perspectives on incorporating additional PA in school, (3) giving children a voice in additional PA participation, (4) the role of the teacher in providing additional PA, (5) taking into account the differences between children in relation to PA participation, and (6) external barriers and facilitators of PA according to children. Figure 1 shows an overview of the themes and subthemes. 

#### 3.2.1. Children’s Motivations for Participating in Additional PA

Generally, children displayed a positive attitude towards additional PA during the school day, stating that it would be “fun” or even “super fun” to be more physically active in school. They also mentioned being dissatisfied with the current amount of PA provided, for example, “*A little more [PE] would be nice*” (girl#7, Focus group (FG) 4). A few children expressed mixed feelings, stating that, “*I think we’re doing enough physical activity in school already*” (girl#2, FG2), or that it would depend on the type of extra PA provided. One boy explained that he did not like PA per se, but that he preferred PA over working on school tasks.

The children mentioned three important reasons for additional PA in school. First, in all focus groups, children discussed the importance of PA for achieving *physical health benefits.* Most children acknowledged that PA is good “*for your body, to stay healthy*” (boy#1, FG5), in particular with regard to physical fitness, maintaining a healthy body weight and strengthening the muscles. Second, children discussed the *emotional benefits* of PA. In the majority of focus groups, children reported that being active is fun and can make you feel better: “*I feel happier and have more energy*” (girl#3, FG9). Likewise, they mentioned that PA can be used as a stress reliever after a test. 

Lastly, in all but three focus groups, children believed that PA could possibly lead to *cognitive benefits,* i.e., improving their ability to focus and/or learn in school: “*You can concentrate better, so it might also help to improve your grades*” (girl#5, FG6). Relatedly, children expressed the need to regularly alternate time spent on academic learning with time being physically active. Children explained that they get distracted, bored, or restless with pent-up energy after long periods of uninterrupted sitting and working on school tasks: “*Then, for example, you have arithmetic, languages, and spelling right after each other, and then you get a little impatient and you really want to move”* (boy#2, FG5). They believed that PA could serve as a break and recharge opportunity: “*You get re-energized*” (girl#5, FG6), which helps them to stay motivated and focused during lessons. However, one girl also stated that being active works in a counterproductive manner and distracts her from work.

#### 3.2.2. Children’s Ideas and Perspectives on Incorporating Additional PA in School

Children had many ideas on how to incorporate additional PA in school, concerning: (1) different PA opportunities, (2) frequency and duration, (3) variation, and (4) location of PA.

*Different opportunities for additional PA:* Children saw opportunities to increase PA time in different parts of the curriculum: during classroom time, recess, physical education (PE), and occasional (outdoor) school activities. In addition, children proposed other ideas, such as the establishment of a child PA committee and possibilities to cut back on academic lesson time in favor of PA. In Table 3, we present all ideas generated by the children, including descriptions and examples. 

*Frequency and duration of PA:* Children did not agree on how long and how often they would like to engage in additional PA opportunities. Concerning the implementation of regular PA breaks, most children agreed on keeping them short, i.e., up to ten minutes. Additionally, in five focus groups children underlined that not too much time should be spent on additional PA, since there has to be enough time left to do school work: “*Yeah because the work still needs to get done*” (girl#2, FG2).

*Variation*: The majority of children emphasized that providing variation is an effective way to make (additional) PA fun, while it also keeps them motivated to participate in PA throughout the school year. Children suggested implementing variation in different ways. First of all, repeating the same activities too often was deemed boring and could negatively impact participation: “*I don’t like tennis or badminton very much and if we have to do that five weeks in a row, I won’t really put much effort in taking part*” (girl#1, FG8). Children discussed the idea of alternating different activities (e.g., tennis one day, soccer the other), and activity types (e.g., collaboration games, ball games, and performing gymnastics exercises). In addition, some children indicated that they would enjoy trying out new activities regularly, for example alternating PE classes with workshops or clinics where they are taught a new sport. Second, children highlighted the importance of regularly replacing and/or expanding the supply of playing equipment in PE and at recess, both because of wear and tear and because “*after a while, it feels like old news*” (boy#4, FG7). Providing variation by purchasing new playing materials and improving the playground facilities (see Table 3) was an important factor for children to facilitate activity during recess throughout the year. 

*Location:* In addition to opportunities for PA in the school building, a reoccurring theme was that children enjoy having activities outside. In all focus groups children came up with one or more suggestions for additional PA outside of the school building, ranging from extending existing “outdoor” time (additional recess and field trips) to moving regular indoor activities outdoors (executing PA breaks in the playground, playing a game in the nearby park).

#### 3.2.3. Giving Children a Voice in Additional PA Participation

Children indicated that they valued having a voice in their PA participation as they emphasized that they would like to have the opportunity to choose the kind of PA that they themselves prefer. For example, one boy said, “*I like it when we get to choose activities ourselves*” (boy#5, FG5). According to the children, this could either be achieved by presenting them a range of options to choose from or by letting them think up their own games or activity program, such as preparing a PE class or, in the case of girls, preparing an academic lesson involving PA. Also, girls specifically mentioned that doing self-invented activities can prevent PA from becoming boring in the long run. However, children also expressed concerns about the school being receptive to their ideas, “*Yeah, but the school always has the last word anyway, no matter what we think*” (boy#1, FG1).

Children also discussed some drawbacks of freedom of choice as some found it difficult to come up with activities themselves. For example, one boy did not prefer so-called “free” PE lessons in which children get to choose their own activities: “*I don’t like that because then I won’t know what to do*” (boy#5, FG7). Moreover, it might actually hamper variation: “*If you’re allowed to choose every time, then I think people will choose the same thing over and over*” (boy#1, FG7). In this respect, two girls suggested providing a box of cards with different games and activities to choose from, which could, for example, be used during recess. Furthermore, some children felt that teachers and supervisors could stimulate them being active by helping them think up fun activities and games. 

#### 3.2.4. The Role of the Teacher in Providing Additional PA

Although some children thought that they would not need much guidance from teachers in executing their ideas for additional PA, others deemed supervision necessary to ensure that PA is safe and enjoyable, in particular to prevent rough play and arguments, and to make sure everyone knows and abides by the rules. According to the children, teachers and supervisors should actively and enthusiastically encourage them to engage in PA: #8: “*Yeah, the teacher should encourage you a little bit, so you won’t just sit on the sidelines and do nothing”.* #7: *“Yeah, like the teacher should say ‘come on, you can do it’*” (girl#8 and #7, FG4). In some cases, children thought it could be motivating when the teacher joins the activities; however, this depended strongly on whether the teacher is considered “fun.” Children were clear that teachers who are considered grumpy or too strict should not join. 

It appeared that the priority given to PA depends strongly on the teacher. Many children mentioned that they have regular PA breaks or extra recess time with certain teachers, but not with others: “*With teacher X, we get to move around every once in a while, but never with teacher Y.*” (girl#6, FG10), or “*Teacher X is more into gardening and being physically active, while teacher Y focuses more on teaching the subjects*” (boy#3, FG5). One boy reported that in his class, children who are behind on schoolwork occasionally have to finish their work at the cost of recess time. In other cases, earlier PA routines implemented by teachers seem to have been forgotten along the way: #6: “*Yes and then we made up these little dances”.* #4: *“Yeah we used to do that every other lesson”.* #6*: “Right, but now the teacher kind of…also because we’ve had the holidays…she kind of forgot about it*” (girls #6 and #4, FG10). 

#### 3.2.5. Taking into Account the Differences between Children in Relation to PA Participation

***Perceived differences in preferences between children:*** Children perceived several differences between children that may influence participation in school-based PA. First, several children mentioned *gender differences*. For example, one boy observed that girls always tend to choose dancing activities (such as Just Dance ^TM^), which, according to him, demotivates boys to be physically active: “*Then we’re not enjoying it, and instead we’ll sit down and not move at all*” (boy#2, FG5). Another boy believed that girls generally prefer talking over being physically active during recess. Both boys and girls mentioned that boys tend to play more roughly, which may be off-putting for girls who want to join. 

Second, children mentioned *age differences.* For example, boys in one focus group perceived the school sports day as “boring” and “not challenging” because they had to do the same activities as the younger children. In addition, several children indicated that children in the lower grades get allotted more recess/playing time, and that the playground equipment is not sufficiently tailored to older children: “*Nowadays, all the sports equipment is made for small children*” (girl#1, FG9). 

Third, children perceived that *differences in skill levels* also influence PA participation, as occurs to this girl when talking about making PE more challenging, “*Yeah, [PE] could be more challenging, but it shouldn’t be made too difficult because some children might not be able to keep up*” (girl#5, FG6). 

Lastly, some children observed differences between *active and non-active* children, “*Some people just prefer sitting on their chair*” (boy#3, FG1). 

***Overcoming the differences between children in relation to PA participation:*** Children discussed that due to above-mentioned differences, it may be difficult to satisfy everyone when it comes to additional PA in school. Although some children did not necessarily experience this as a problem—“*Yeah but there will always be something that somebody doesn’t like*” (girl#5, FG2)—many others emphasized the importance of “*keeping it fun for everybody*” (girl#3, FG2). In most focus groups, children discussed efforts that should be made to make sure that all children (want to) participate. Without necessarily agreeing on one best option, they proposed several potential strategies to take into account children’s different needs and preferences. 

First of all, according to most children, participation can be encouraged by either *allowing children to choose* their own activities, or to choose from a range of options which increases the chance of everyone finding something that fits their interest. In this respect, some children suggested organizing separate activities and/or providing different equipment for girls and boys and for children with diverse skill levels (see also “Providing different difficulty levels in PE” in Table 3). To ensure a fair selection process, children proposed to let everyone take turns in choosing PA, rotate activities often, and/or to decide on activities by voting or drawing lots. 

Second, a couple of children suggested *making participation mandatory* for all children, regardless of them actually liking the activities. Some even proposed a penalty for non-participation such as staying after school. However, many others preferred to always have the possibility to opt-out because they did not feel that participating is fun if you do not like the activities, which might lead to half-hearted participation at best, and at worst to an unsafe PA environment: “*I would only let children do it who enjoy [participating in PA] because if some of them don’t enjoy it they might ruin it for the others*” (girl#2, FG8).

A third line of thinking in some focus groups with girls concerned employing *positive reinforcement.* Girls suggested that children or teachers could try to encourage hesitant children to join in games during recess or offer them a reward as an incentive to participate: “*Maybe they can be allowed to do something else afterwards, like drawing*” (girl#6, FG4). In one focus group, boys indicated that it would be important that teachers provide a rationale for (additional) PA: #4: “*Well, teachers should stimulate [PA] more and also explain the purpose of it […] because some children ask themselves, why are we doing this?*”. #1: “*Yeah, why should you be active?*” (boys#4 and #1, FG3). Finally, a few girls pointed at the benefits of *habituation*: “*No you should just let him participate and in the end he might learn to like it because he’s done it so often*” (girl#5, FG6). Therefore, they considered it important that children are willing try out new activities that are chosen by others. 

#### 3.2.6. External Barriers and Facilitators of PA According to Children

Lastly, children mentioned several external barriers and facilitators that may influence the implementation of (additional) PA in school.

***Weather:*** The weather was mentioned as an important determinant of (additional) PA time. One boy indicated that they are often kept inside during recess when it rains: “*Usually when it rains we don’t go outside but stay in and watch a video or film or something*” (boy#5, FG7). Another girl remarked that the school sports day had recently been cancelled twice due to bad weather. As a solution, two boys proposed to catch up lost recess time later, and to buy adequate outdoor boots for everyone. Conversely, children mentioned that when the weather is good, they sometimes get extra PA time outdoors, such as playing games in the nearby park. 

***Current school policies****:* Children also gave examples of school policies that impede being active in school. For instance, they mentioned that some areas that offer opportunities to be physically active, such as the bushes, the lawn, and access to equipment in the shed, are restricted to them. In one focus group (FG7) children mentioned that they do not have recess on Wednesday since they already have PE that day. 

***Lack of space and resources:*** Children recognized and discussed that carrying out some of their ideas for extra PA would require considerable planning and organization because of limited space and resources in the school. Therefore, they suggested to keep group sizes manageable, implement activity timetables (e.g., a rotation system to use the Wii gaming console), and build multi-functional exercise facilities, such as a soccer field that serves as an ice-skating rink in winter. 

***Costs:*** The children were also very conscious of the monetary costs connected to their PA ideas, and therefore designated some ideas as not feasible in advance, e.g., weekly school outings that cost money or buying gaming consoles for everyone. As a solution to financial barriers, children suggested organizing a charity run or fundraising activity to raise money, for example, to buy new playground equipment or materials.

## 4. Discussion

To the best of our knowledge, this is the first study that comprehensively explored the views and needs of 10–13-year-old children to increase PA opportunities in the broader primary school setting. Children provided various ideas for additional PA and discussed important factors that need to be taken into account when developing feasible, enjoyable, and sustainable school-based PA programs. In line with the socio-ecological model [63], children in our study mentioned factors related to opportunities to be physically active in school on multiple levels (i.e., individual level, social level, physical environment, and school policy level). We will discuss the main findings of our study with a focus on their novelty and their implications for research and school practice.

### 4.1. Motivations for Additional PA in School

Despite the fact that children with different activity levels participated in each focus group, almost all children in our study were enthusiastic about additional PA in school, which is in line with earlier research showing that many children and young people generally enjoy participating in PA [64], and want to be more active [65]. Reasons for this appeal are similar to findings of other studies: children considered PA important because it is fun, provides enjoyment and/or feelings of happiness [15,28,64,66,67], and because it helps to stay fit and healthy [15,28,66]. A novel finding of our study, expanding on the existing literature, was the importance of the perceived cognitive benefits of PA, specifically in the school setting. Children indicated that PA helps them to increase their motivation and focus, which was, according to them, particularly important given the long and uninterrupted bouts of sitting and/or working on school tasks during a school day. Children’s expressed motivations for and needs to be physically active during school time reflect the importance and relevance of increasing PA opportunities in primary schools. In line with our findings, teachers have also reported that children have a need to move regularly during the school day to restore and increase their attention [23,68]. Moreover, these perceived “cognitive” benefits have been shown a key argument in teachers’ willingness to implement additional PA in school [23,24,25]. 

To date, however, recommendations to increase school-based PA have often targeted increasing knowledge on the long-term health benefits of PA [15]. It might be that in schools, strategies focusing on the “here and now” benefits of PA, such as enjoyment, increased attention, and a good atmosphere in the classroom, may connect better to children’s and teachers’ day-to-day experiences and motivations and therefore be a more promising avenue to integrate more PA in schools. In this respect we must note that the scientific evidence for the effects of PA on cognitive outcomes, as measured by objective and standardized measurement instruments, is still inconclusive [69,70]. On the other hand, two recent systematic reviews found that PA breaks can improve children’s classroom behaviour [70] and school engagement [71], which seem more closely related to the PA benefits that teachers and children experience. To gain more insight in these “practice-based” effects of PA, we recommend including children’s and teachers’ experiences as additional outcome measures in future research on school-based PA interventions.

Although almost all children acknowledged the importance of PA and showed a positive attitude towards additional PA in school, there appeared to be a gap between “willing” and “having the opportunity” to participate in (additional) PA in school. In line with the findings of a systematic review of Morton et al (2016) focusing on adolescents [72], children in our study specifically mentioned the importance of factors on the social level, i.e., teacher support, to address this gap. For example, children indicated that they noticed vast differences between teachers with respect to what PA they offer, or their willingness to implement extra PA. Therefore, teachers’ priorities and affinities with PA seem to have a strong influence on whether or not children engage in (additional) PA at school. Thus, our results suggest that an important step to create more opportunities for children to be active in school is to overcome barriers at a teacher level. However, teachers’ priorities are likely shaped by organizational and policy level factors. It is, for example, well known that teachers feel time constraints due to an overcrowded academic curriculum and pressures to reach academic targets [23,33,41,73]. While increasing children’s academic performance is an important core goal of school and a responsibility of teachers, the current focus on academic performance subsequently reduces the willingness to increase time for PA during school hours. Our results indicate that these pressures also extend to the children. In more than half of the focus groups, children emphasized that not too much time should be spent on additional PA at cost of time for school tasks. However, compelling evidence shows that spending additional time on PA at the cost of time for learning has no adverse effects on children’s academic performance [69,74,75]. Therefore, we recommend informing policy makers, schools, teachers, and children about these findings in order to address and take away their concerns and to create more support for additional PA in schools. Furthermore, these insights might stimulate schools to adapt some of the policies that are clearly within their realm of control, such as not withholding recess, to give room for more PA. 

### 4.2. Sustainable Participation in PA

Children mentioned several factors that are, according to them, important to stay motivated to participate in PA. We will now highlight the major themes that emerged. Children stressed the importance of involving them and giving them voices when it comes to additional PA in school. Children particularly recognized motivational benefits of student participation in relation to durable PA interventions, as they indicated that involving them in choosing and inventing activities will motivate them to sustain in PA participation in the long run. This finding can be explained using the self-determination theory, which postulates that three basic psychological needs (i.e., competence, autonomy, and relatedness) need to be fulfilled in order to have children intrinsically motivated to engage and sustain in a health behaviour such as PA [42,76]. Within this theory, the need for autonomy is a particularly important factor to attain high levels of intrinsic motivation [42]. However, our study also revealed that children questioned whether the school would actually be open to implementing their ideas, which indicates that it is currently not common to involve children in choosing and designing school-based activities.

On the other hand, children also indicated a need for guidance in their PA choices as they suggested that teachers can help them in choosing or designing PA activities. This aligns with the finding that, to successfully implement educational innovations and school health promotion programs, co-creation with all stakeholders is recommended, leading to increased ownership of the intervention in both students and teachers [22,77,78]. Moreover, teacher support and encouragement appeared important to keep children participating in school-based PA over time. 

Our results indicate that 10- to 13-year-old children are very capable of inventing solutions to perceived barriers (e.g., sharing equipment, fundraising to collect money, using timetables, catch up on lost PA time). Hence, it seems of considerable value to involve children actively and make “additional PA in school” a shared project of teachers and children. One specific way in which to increase student participation in school-based PA could be establishing a child-PA-committee, as suggested by some boys in our study. Lastly, children considered variation not only important to make PA enjoyable (see Section 4.3), but also considered it a major factor in keeping them interested in participating in additional PA in the long run. In particular, they highlighted the importance of variation in terms of (types of) activities and PA equipment, which indicates that it is important to adapt and change school-based PA programmes regularly. 

### 4.3. General PA Promotion versus School-Based PA

In line with earlier research on PA promotion in general (i.e., not specifically in the school setting), children in our study reported several factors that are important to make school-based PA enjoyable, such as variation [27,37,43,66,79], choice in PA activities [28,33,37,42,66], providing encouragement [73,79] and new activities [27,33], supervision of teachers [43,80], and using technology [27,37]. In contrast to earlier studies [72], children in our study did not emphasize the importance of competitive elements in PA, which might be due to differences in age of the participants (preadolescents in our study versus adolescents in Reference [72]) or the setting of PA. Also, being physically active with friends has previously been reported by children as an important factor to make PA fun [27,33,79], but was not emphasized by the children in our study. This contrast between our and previous studies might be explained by the different settings of PA promotion, i.e., in school versus PA in general. Similar to earlier studies, children’s perceived barriers towards school-based PA included bad weather [36,79] and lack of time to fully utilize PE [28,29]. Furthermore, in line with teachers and principals [23,81], the current school policies, lack of space and resources, and financial constraints were also recognized by the children as barriers towards school-based PA. Overall, we can conclude that most facilitating factors and barriers in general PA promotion also apply to PA promotion specifically in the school setting. Expanding on the existing literature on general PA promotion, our study provides a comprehensive and concrete overview of children’s voices with regard to preferred and enjoyable PA opportunities in school (see Table 3). This overview is of great value for future research and practice, as it can be used as a basis for developing school-based PA programmes that are appropriate for and attractive to 10-to-13-years old children. 

### 4.4. Feasible Opportunities for Additional PA

Many of the children’s suggestions centered around expanding time of the existing PA curriculum, such as increasing the duration and frequency of PE, recess, and occasional activities (e.g., sports day, school outings). These findings confirm the results of earlier studies and reflect the need of children to be more physically active during school time [37,73,82]. However, if we look at the myriad of barriers that schools currently experience, these forms of additional PA are difficult to realize in daily school practice due to time constraints and the financial and material resources that are needed, e.g., hiring PA professionals, acquiring new (technological) play equipment, and adapting playground facilities [41,73]. This was also recognized by children in our study, whom, for example, worried about too little time remaining to finish school work. In the Netherlands, schools are accountable for reaching academic targets set by the government [83], and receive a budget from the government that can be used at the school’s discretion to hire staff and for maintenance of the school building [52]. In this sense, schools have some flexibility in deciding to spend (part of these) resources on facilitating more PA. However, to realize substantial extensions of the current PA curriculum, considerable investments and shifts in priorities of the school board or even the political level (such as installing legal provisions prescribing a minimum amount of PE in schools) seem necessary [81]. 

Additionally, children had various ideas for additional PA in school that seem relatively easy to realize in daily school practice. For instance, children indicated that more effective PA time could be achieved during current PE lessons, for example by playing music to stimulate movement during waiting times, by providing short activities in between exercises/games, or by providing exercises with different intensity levels for children that need more of a challenge. Likewise, providing children with activities during recess could also stimulate them to be (more) physically active. These suggestions all appear within the control of a (PE) teacher to realize and can therefore be relatively easily implemented. In the case of the Netherlands, PE is often taught by classroom teachers instead of PE specialists, which could potentially hamper the delivery of quality PE [84]. Investing in a physical education specialist or PE support for generalist instructors therefore seems prudent. 

Another suggested form of feasible additional PA was PA in the classroom, either in the form of short exercise breaks (suggested in all focus groups) or physically active academic lessons (only suggested by girls). Children preferred to have the exercise breaks in between lessons, which aligns with their need to regularly alternate working on school tasks with being active. Children’s suggestions for classroom-based PA are in line with forms of PA that teachers and principals consider most feasible in school, i.e., short exercise breaks of 5 to 10 minutes [23,85] or active academic lessons [86]. Both PE-related interventions and interventions that involved activity breaks have been proven to be an effective way to increase children’s PA in the school setting [87]. Finally, in line with the findings of Hyndman [37], children often suggested PA in the form of games or PA that included a game component. Earlier intervention research on recess activities shows that providing a game-based curriculum [88,89] and game equipment [90] can successfully increase children’s overall PA levels. Playing games and incorporating “game” features or equipment within exercise bouts might therefore be an effective way to increase PA in children.

### 4.5. Tailored PA Programs

Also, worthwhile reporting is the apparent contradiction between children’s need for “tailored PA” and “including everyone in PA.” On the one hand, children indicated the importance of taking into account their individual needs and preferences, while on the other hand they emphasized that efforts must be made to make sure that all children can participate in additional PA in school. Related to the first, earlier studies have repeatedly recommended developing individually tailored PA programs [15,27,41,43,65]. However, these recommendations are rarely followed-up since the development of such PA programs is very difficult due to considerable heterogeneity in school populations and corresponding logistical challenges [41,65]. Hence, most PA programs are still of a “one size fits all” type. Our results indicate that it is important to find a middle way in this paradox i.e., finding solutions to meet individual preferences within a group setting. 

Although children emphasized the importance of taking into account individual preferences, another new finding of this study is that they also indicated that it is fine to perform PA activities that they consider less fun, as long as everyone’s preferences are represented once in a while. As such, it seems important for future research to invest more in active engagement/participation of children when developing PA programs [16,77,78]. It is important that children can discuss the content of the PA program and find compromises when preferences differ. Co-creation of PA promotion programs become more prevalent [78] and future effectiveness trials should be conducted to gain insight regarding whether co-created PA promotion programs are more effective in improving physical activity and subsequently health and school-related outcomes.

### 4.6. Strengths and Limitations

Our study has several strengths. First, we used children’s views as the starting point and children were not restricted by researchers’ preconceived notions, resulting in an in-depth and comprehensive exploration of children’s perceptions of additional PA in the primary school setting. Second, we included a task-based activity, resulting in a broad representation of different viewpoints and opinions. Third, we included schools in both regional and urban areas of the Netherlands, and made efforts to include children with different PA levels in each focus group to avoid selection bias. Fourth, by including our interview guide and an extensive audit trail, we provide trustworthiness and transparency in the data collection and analysis process. 

Our study also has some limitations. First, due to the nature of qualitative research, our findings may have limited generalizability, for example, to other school systems and to children with other cultural backgrounds (most children were of Dutch origin). Moreover, we have to keep in mind that each school has its own context, which influences children’s ideas regarding PA opportunities in their particular school. Second, although we included children with different activity levels, it is possible that more children who were already enthusiastic about PA signed up to participate. A third potential limitation is that we started the data analysis after data collection was concluded, which prevented ad hoc adjustments to the interview protocol. However, we pilot-tested the interview guide beforehand. Fourth, one focus group included eight girls and was difficult to moderate. This could have inhibited some girls to express their opinions. Lastly, we aimed to collaborate with two interns of an external institution, but unfortunately, we noticed that they did not follow the interview protocol. Therefore, we decided not to use data from these two focus groups. Although the interns wrote their internship report on this data, we could not fully avoid research waste. Nevertheless, we reached saturation with the available focus groups.

## 5. Conclusions

In general, primary schoolchildren who participated in this focus group study would welcome additional PA opportunities in school and expressed a desire to be more physically active during the school day. Schools and researchers should capitalize on this enthusiasm when developing PA programs, while the child-perceived beneficial practice-based effects of additional PA, such as restored attention in the classroom, warrant further investigation. Children in this study suggested various ways to increase PA in school, of which finding ways to improve effective PE time and providing short activity breaks in between lessons seem relatively easy to implement in daily school practice. Future research could provide further insight into whether including the child-identified suggestions are indeed effective in structurally raising their PA levels. Furthermore, children perceived choice and variation as important for keeping the PA options attractive in the long term and identified teachers as both a key barrier and facilitator of PA participation. We therefore recommend actively involving children in efforts to increase school-based PA and to make “additional PA in school” a shared project of teachers and students. Overall, our study provides a comprehensive overview of children’s voices regarding additional PA in school, which could be used to inform the development of future PA interventions aimed at increasing the activity levels of children in primary school. To ensure relevance to local contexts, it is important that these strategies include the involvement of children, teachers, and other key stakeholders. 

## Figures and Tables

**Figure 1 ijerph-15-02713-f001:**
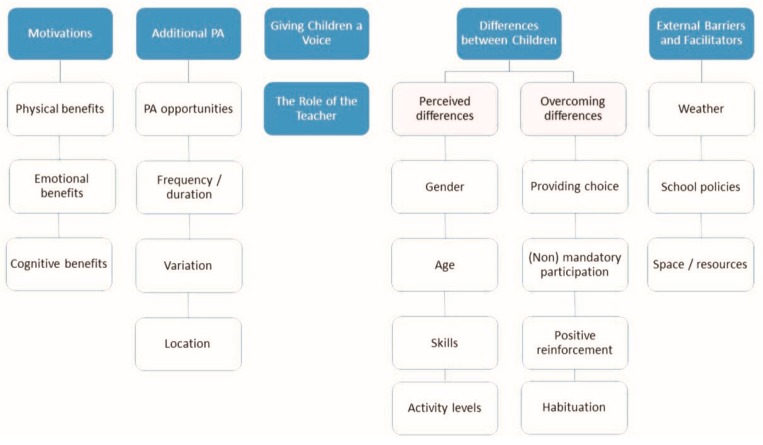
An overview of all themes and subthemes resulting from the analysis. PA: physical activity.

**Table 1 ijerph-15-02713-t001:** Semi-structured focus group discussion guide.

Question Number	Focus Group Question
Warm-up question 1	What do you think we mean by “physical activity”?
Warm-up question 2	What kind of physical activity do you currently do in school?
Question 1	Let’s say you start being more physically active in the classroom or in school, how would you feel about that?
Question 2 *	If you could tell the school your ideas about how to be more physically active during the school day, what would you say?
Question 3	What do you need to be more physically active in school?
Question 4	What if the school decided to let you be more physically active throughout the entire year. How could we make sure physical activity keeps being fun and interesting?

* Including a task-based activity, i.e., writing individual ideas on sticky notes.

**Table 2 ijerph-15-02713-t002:** Audit trail of the data analysis, following the steps of Braun and Clarke [59].

Step		Description			Examples
1. Familiarizing yourself with the data		V.B. and research assistant transcribed focus group data verbatim. E.V. and V.B. read all transcripts and discussed initial ideas and interesting features.			Initial interesting features: Children seemed to have many ideas regarding PE, active games, recess activities, and activities outside the school building.
2. Generating initial codes		E.V. and V.B. open coded two transcripts independently, selecting relevant text fragments, and ascribing initial codes. After each transcript, E.V. and V.B. compared their work and discussed until consensus was reached. E.V. coded the remaining transcripts. Each time E.V. had coded either one or two transcripts, V.B. checked the manuscript(s) and supplemented relevant text fragments and assigned codes. Discrepancies were discussed until consensus was reached. This process was repeated until E.V. and V.B. concluded that coding the transcript yielded no significant new codes in relation to the previously coded transcripts. This was the case after coding four focus groups with boys and five focus groups with girls.			Initial code examples: “PA barrier: weather”, “PA facilitator: teacher”, “Resources: playing equipment”, and “PA motivation: health”. The coders discussed whether the text fragment “*I would like to play more sports in school*” should be coded as “Need to be more active” or “Need for more PE”. The coders discussed whether children meant the same thing when they indicated that they prefer more “Workshops” or “Clinics”.
3. Searching for themes		E.V. re-read all coded data, comparing the coded extracts to the assigned code names. Similar codes were grouped together into initial (sub)categories. The collated text fragments of (sub)categories were read and re-read to identify potential overarching themes.			The codes “Doing the same activities” and “Alternate location of PA” were grouped together in the subtheme “Variation” and theme “Characteristics of additional PA”.
4. Reviewing themes		E.V. formulated a preliminary map of the main (sub)themes and how they related to each other. The coherence and distinctness of the themes and subcategories were first discussed and revised together with V.B., and subsequently within the larger research team (E.V., V.B., A.S., and M.C.).			Initial themes that were identified: “Children’s motivations”, “Characteristics of additional PA”, “Influences on enjoyable PA” (Choice, Personal preferences, Inclusion, and Supervision), “External barriers and facilitators”.
5. Defining and naming themes		Going back and forth between all data, E.V. refined the content of each (sub)theme, collated significant quotes and wrote a first draft of the results. E.V. and V.B. reflected on the first draft of the results in detail until consensus about clear definitions of the (sub)themes was reached.			The theme “Characteristics of additional PA” was revised into “Additional PA according to children” with subthemes “Variation”, “Location”, etc. Some of the subthemes were revised into a main theme; for example, subtheme “Inclusion” was revised into main theme: “Taking into account the differences between children” with subthemes “Perceived differences” and “Overcoming differences”.
6. Producing the report		E.V. refined and completed the report of the data analysis with input from V.B., A.S., M.C., and R.G.			

Note: E.V.: Eline Vos; V.B.: Vera van den Berg; A.S.: Amika Singh; M.C.: Mai Chin A Paw; R.G.: Renate de Groot; PE: physical education; PA: physical activity.

**Table 3 ijerph-15-02713-t003:** Primary schoolchildren’s suggestions for integrating additional physical activity (PA) in school.

	Mentioned in Focus Group *	Description	Examples
Classroom time:			
PA breaks	B1, G2, B3, G4, B5, G6, B7, G8, G9	Short PA breaks (up to ten minutes) in between tasks, usually involving a game component. Either in the classroom itself or in/around the school property	Various games that require moving (relay, hide and seek, Twister ^TM^); dancing breaks, (e.g., Just Dance ^TM^); short exercise activities (e.g., plank exercise, jumping jacks, squads, running around your chair, running a lap around the school)
Technology-based PA breaks	B1, B3, B5, G6, G8	PA games involving technology	Active gaming (Nintendo Wii ^TM^); Virtual reality games (Oculus Rift ^TM^)
Incorporating PA in academic lessons	G4, G6, G9	Movement integration (1) related to, or (2) unrelated to the content of the academic lesson	(1) Having to run to the right answer of addition problems in math lesson; answering language or math questions while tackling an obstacle course; (2) using bicycle desks
Physical Education:			
Extra PE time	B1, G2, B3, G4, B5, G6, B7, G8, G9	Increasing frequency and/or duration of PE	Children had differing opinions on how often or how long PE should be taught
Staying active in PE:			
*Playing music in PE*	G8	Playing music helps you be more active as you automatically start moving to the music in between exercises.	“*When you have to wait in line you can see everyone moving a little to the music*” (girl#2)
*Providing different PA levels*	B3, G4, G6	Providing or increasing different levels of PA to accommodate children who prefer (1) more challenge and/or (2) more intensive activities	(1) Challenging activities, such as athletics, gymnastics jumps, obstacle course or rope climbing; (2) activities or sports that make you tired such as running or cardio
*Increasing effective PE time*	B1, G2, B5, G6, G9	Increasing the time children are actually physically active in PE	Limiting waiting time in between exercises and time spent on travelling, changing clothes, setting up, and putting away equipment; providing short alternative activities if you have to wait for a longer period of time (e.g., during a Dutch variant of softball)
Recess:			
Extra recess	B1, G2, B3, G4, B5, G6, B7, G8, G9	Increasing frequency and/or duration of recess	Children had differing opinions on how often or how long extra recess should be implemented
Improving the playground	G4, B5, G6, B7, G8, G9	Making physical changes to the playground and area around the school to encourage more (active) playing	Updating or installing new playground equipment, such as trampolines, table tennis table, adventure/obstacle course, climbing frame, soccer field, water slide, swing set, indoor playground
Occasional activities:			
Extra school sports days	G4, B5, G6, B7	Increasing frequency of school sports days, where children play different kinds of sports and games for the entire or a part of the day	Different games and sport activities (new and familiar), obstacle course, discus throw, long-jump
Extra clinics/workshops	B1, B3, G4, B5, G6, G9	Increasing frequency of inviting a sports expert to teach children new skills/sports in addition to PE	(Rescue) swimming, street dance, hip hop, baseball, learning new/unfamiliar sports
Extra stage performances	B5, G6, G9	Increasing frequency of stage performances where children prepare an act and perform it in front of all children	Different acts, such as drama, singing, and dancing
Extra active field trips	B1, B3, G4, G6, B7, G9	Increasing frequency of active field trips	Going to the beach or to the park, camp, cycling, paintball, indoor playground, laser gaming, swimming, climbing, amusement park
Using bicycle as transportation	B3, G8, G9	Using bicycles more often or as standard mode of transportation to and from field trips	Cycling to the beach, the swimming pool, the park or to camp
Other:			
PA committee	B1	Establishing a child PA committee that organizes fun and interesting activities for the children in school	Preparing PE class or physical activity for classmates and/or children in other grades
Cutting back on academic lesson time	B1, G9	Cutting back on academic lesson time to spend more time on PA	Letting out school early to do PA on Fridays; shortening lessons to provide more time for activity breaks; giving children one free period per day to do a physical activity of their own choice

G = focus group with girls; B = focus group with boys.

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
