# Peer review of "Untapped Resources: 10- to 13-Year-Old Primary Schoolchildren’s Views on Additional Physical Activity in the School Setting: A Focus Group Study"

_ijerph, 2018, doi:10.3390/ijerph15122713_

Round 1

Reviewer 1 Report

This research investigation is a worthwhile endeavour. This paper is well prepared and has the potential to make a valuable contribution to the literature. Below, I am providing several comments and suggestions that the authors may choose to use in revising their manuscript.

Introduction:

Given the focus of the paper, I suggest strengthening the rationale for examining children voice / perspectives.

The authors may find this paper useful:

Jacquez, F., Vaughn, L.M., & Wagner, E. (2013). Youth as partners, participants or passive recipients: A review of children and adolescents in community-based participatory research (CBPR). American Journal of Community Psychology, 51(1-2), 176-189.

In the introduction or the methodology, I suggest that the authors provide more context for PA in schools in the Netherlands – are there any policies and guidelines about curriculum time allocation for PE, who is teaching PE, recess time, are schools autonomous in terms of making relevant decisions, etc. The following paper that focuses on PA policies in Australian schools may be helpful with this task:

Stylianou, M., & Walker, J. L. (2018). An assessment of Australian school physical activity and nutrition policies. Australian and New Zealand journal of public health, 42(1), 16-21.

Given previous research, are there any frameworks that can help position the study? Even if no frameworks are used in the introduction, some theories may be useful to refer to in the discussion. For example, autonomy (choice), which is emphasised in the paper, is a main component of the Self-Determination theory that focuses on motivation to engage in particular activities. Another example is the ecological model that involves multiple levels of factors that influence behaviour – many of which are mentioned in this paper.

Methods:

Line 95: change ‘characteristic’ to ‘characteristics’ (plural)

Line 119: what do you mean exactly by ‘did not structurally provide PA programs’

Focus group composition – can you please provide more information about how many students from each level of PA behaviour ended up participating?

Discussion:

Lines 417 – 419: Is this supported by the results of the current study? Please clarify / explain.

Lines 435 – 442: How are the results of these reviews related to your findings? Do they support or contradict them? Or, are you using these findings here to demonstrate the importance of student voice? Please explain. You touch on motivation benefits in the following paragraph but make no connections with the rest of the findings of these reviews.

Line 454: I suggest using ‘establishing a child-PA-committee’ (rather than installing).

Lines 471 – 482: Thinking of this paragraph, are you suggesting that despite the importance of giving children voice, they sometimes lack the necessary knowledge to know whether they are realistic?

Line 524: This sentence needs to be revised – do you mean ‘trials should be conducted…’

Given the focus of this study on ‘sustainability’, I suggest that more emphasis is given on this aspect in the discussion.

Author Response

This research investigation is a worthwhile endeavour. This paper is well prepared and has the potential to make a valuable contribution to the literature. Below, I am providing several comments and suggestions that the authors may choose to use in revising their manuscript.

We thank the reviewer for his/her compliments, comments and suggestions for further improvement of our manuscript. We considered all comments and suggestions and highlighted the adaptations in the revised manuscript in yellow.

Introduction

Given the focus of the paper, I suggest strengthening the rationale for examining children voice / perspectives.

The authors may find this paper useful:

Jacquez, F., Vaughn, L.M., & Wagner, E. (2013). Youth as partners, participants or passive recipients: A review of children and adolescents in community-based participatory research (CBPR). American Journal of Community Psychology, 51(1-2), 176-189.

We thank the reviewer for this valuable paper suggestion. The reviewer’s suggestion to strengthen our rationale for examining children’s perspectives is in line with comments of reviewer #2 and #3. We have therefore included additional information in multiple parts of our introduction section (see highlighted text below), including the suggested reference of Jacquez et al. (2013).

Introduction, line 52-71

However, several systematic reviews concluded that school-based PA interventions often have only limited or short-term effects on children’s overall PA levels [13, 14]. A possible explanation for this finding is that the evaluated interventions are commonly developed with limited involvement of school staff and children and may therefore be insufficiently tailored to their specific circumstances, needs and wishes [15-17]. This could result in the development of inappropriate interventions resulting in sub-optimal implementation of and participation in these interventions, limiting their effectiveness [16, 18, 19]. In order to ensure that school-based PA interventions will succeed in daily school practice, they need to be relevant to local school contexts and appropriate for and attractive to the intended users (i.e. children and school staff) [15-17, 19-22]. A recent systematic review concluded that involving students in the development and implementation of school-based health activities, has positive effects on students’ motivation and attitude towards the activities, health-related outcomes, social interactions, as well as the school’s culture, climate, rules and policies [23]. Moreover, an earlier systematic review by Jacquez et al (2013) has shown that involving children and adolescents in research projects can lead to increased quality of research and higher chances of translating research outcomes into action in daily practice [17].

In the past years, several qualitative studies have investigated the perspectives of teachers and school administrators regarding primary school-based PA interventions [e.g. 24-27]. These studies provided valuable information on facilitators and barriers, mostly regarding the organisation and execution of PA. However, children are the ones who are expected to actually participate in the intervention, and it is very likely that their perspectives differ from the views of adult school staff.

Line 86-100
There is currently a lack of qualitative studies that focus specifically on PA opportunities in the primary school setting, and in particular involving the critical age group of 10-13 years, where, as mentioned above, PA tends to decline [7, 8]. Moreover, the qualitative studies investigating additional PA opportunities that have been conducted in the primary school setting often have a narrow focus, determined in advance by the researchers, such as PA promotion in recess [e.g. 35-37], or adapting the school physical environment [e.g. 38-40]. For example, a focus group study of Hyndman (2016) investigated what kind of school PA facilities 10-13-year-old children find appealing (e.g. a variety of recreational, sporting and adventure facilities). However, this study was limited to features of the physical environment and did not consider other ways of enhancing school-based PA. As previous studies focused on certain predefined aspects of the school day, it remains unclear how exactly PA opportunities can be increased within the broader primary school setting according to children. In particular, knowledge is lacking on what kind of activities would be appealing to children, as well as what they need to successfully engage in school-based PA. Therefore, a comprehensive investigation of children’s thoughts and ideas concerning PA promotion within the primary school setting in general is required.

Line 112-121
In sum, for the development of attractive and effective PA interventions in primary school, more knowledge of the perspectives, needs and wishes of children is needed. To the best of our knowledge, no qualitative studies have explicitly and comprehensively explored the multi-faceted ideas of 10-13-year-old children to increase PA in the primary school setting. Therefore, the aim of our study was to 1) explore 10-13-year-old primary schoolchildren’s perspectives on incorporating additional PA into the school day, in particular what practical ideas children have to be more physically active in school and what they think is important to successfully implement these ideas; and 2) to investigate children’s views on how to keep the proposed activities and ideas interesting and fun for a longer period of time. The results of this study can inform both research and practice regarding the development and implementation of future PA interventions aimed at 10-13-year-old children.

In the introduction or the methodology, I suggest that the authors provide more context for PA in schools in the Netherlands – are there any policies and guidelines about curriculum time allocation for PE, who is teaching PE, recess time, are schools autonomous in terms of making relevant decisions, etc. The following paper that focuses on PA policies in Australian schools may be helpful with this task:

· Stylianou, M., & Walker, J. L. (2018). An assessment of Australian school physical activity and nutrition policies. Australian and New Zealand journal of public health, 42(1), 16-21.

We thank the reviewer for this useful suggestion and helpful paper. We have added an additional subsection (2.2) to the methods section in which information about the Dutch primary school system is provided.

Methods line 132-144
2.2. PA in the Dutch primary school system

In the Netherlands, children aged 4 to 12 attend primary school (2 years of pre-school followed by grades 1 to 6). On average, children participate in an hour and a half of PE per week. There are no legal provisions issued by the government on the amount of PE that should be provided, but the guideline of the Ministry of Education, Culture and Science prescribes at least two to three lessons of PE per week. PE should either be taught by classroom teachers with a qualification for PE, or a PE specialist [51]. Schools are free choose their own recess hours, e.g. a short break in the morning and a longer one for lunch break, or two breaks of the same length [52]. Furthermore, schools receive a budget issued by the government that they can spend at their discretion on facilities or personnel, such as a PE teacher [53]. In addition to the regular PA curriculum, there are some national/municipal initiatives (e.g. Healthy Schools, Jump-In, PLAYgrounds, JOGG) that promote extra physical activity in schools, e.g. by providing PA interventions, school (playground) adaptations, or external coaches who organise PA activities during recess or after school hours.

Given previous research, are there any frameworks that can help position the study? Even if no frameworks are used in the introduction, some theories may be useful to refer to in the discussion. For example, autonomy (choice), which is emphasised in the paper, is a main component of the Self-Determination theory that focuses on motivation to engage in particular activities. Another example is the ecological model that involves multiple levels of factors that influence behaviour – many of which are mentioned in this paper.

The reviewer poses a very valid question. To stay as close as possible to the children’s voices concerning school-based PA we chose to employ an inductive analysis approach in this article, i.e. we opted to let the themes emerge from the data instead of using a framework/theory to analyse the data upfront. According to the reviewer’s suggestion, we have related our results to existing theories on PA behaviour in the revised version of our discussion section.

Line 436-443

To the best of our knowledge, this is the first study that comprehensively explored the views and needs of 10-13-year-old children to increase PA opportunities in the broader primary school setting. Children provided various ideas for additional PA and discussed important factors that need to be taken into account when developing feasible, enjoyable and sustainable school-based PA programs. In line with the socio-ecological model [63], children in our study mentioned factors related to opportunities to be physically active in school on multiple levels (i.e. individual level, social level, physical environment and school policy level). We will discuss the main findings of our study with a focus on their novelty and their implications for research and school practice.

Line 472-484

Although almost all children acknowledged the importance of PA and showed a positive attitude towards additional PA in school, there appeared to be a gap between ‘willing’ and ‘having the opportunity’ to participate in (additional) PA in school. In line with the findings of a systematic review of Morton et al (2016) focussing on adolescents [72], children in our study specifically mentioned the importance of factors on the social level, i.e. teacher support, to address this gap. For example, children indicated that they noticed vast differences between teachers with respect to what PA they offer, or their willingness to implement extra PA. Therefore, teachers’ priorities and affinities with PA seem to have a strong influence on whether or not children engage in (additional) PA at school. Thus, our results suggest that an important step to create more opportunities for children to be active in school is to overcome barriers on a teacher level. However, teachers priorities are likely shaped by organisational and policy level factors. It is, for example, well known that teachers feel time constraints due to an overcrowded academic curriculum and pressures to reach academic targets [23, 33, 41, 73].

Line 495-508

4.2. Sustainable participation in PA

Children mentioned several factors that are, according to them, important to stay motivated to participate in PA. We will now highlight the major themes that emerged. Children stressed the importance of involving them and giving them voices when it comes to additional PA in school. Children particularly recognised motivational benefits of student participation in relation to durable PA interventions, as they indicated that involving them in choosing and inventing activities will motivate them to sustain in PA participation in the longer run. This finding can be explained using the self-determination theory which postulates that three basic psychological needs (i.e. competence, autonomy and relatedness) need to be fulfilled in order to have children intrinsically motivated to engage and sustain in a health behaviour such as PA [42, 76]. Within this theory, the need for autonomy is a particularly important factor to attain high levels of intrinsic motivation [42]. However, our study also revealed that children questioned whether the school would actually be open to implementing their ideas, which indicates that it is currently not common to involve children in choosing and designing school-based activities.

Methods

· Line 95: change ‘characteristic’ to ‘characteristics’ (plural)

We have changed ‘characteristic’ to ‘characteristics’.

· Line 119: what do you mean exactly by ‘did not structurally provide PA programs’

We apologise for being unclear. In the Netherlands, there are some national/municipal initiatives (e.g. Healthy Schools, Jump-In, PLAYgrounds, JOGG) that aim to promote physical activity in schools, e.g. by providing PA interventions, school (playground) adaptations, or external coaches who organise PA activities during recess or after school hours. These initiatives are used in addition to or as part of the regular physical education curriculum and recess, and are based on previous research. We excluded schools that provided these programs, because it is likely that the ideas of children who participate in such PA interventions are influenced by their PA experiences. Moreover, the majority of schools in the Netherlands do not participate in these initiatives, and the views of these children may therefore differ from the views of children attending ‘regular’ primary schools. As mentioned before, we were looking for PA ideas from the perspectives of children, i.e. with a “bottom-up” approach, and therefore tried to avoid any “top-down” influences. As suggested by the reviewer we added an additional paragraph on PA in the Dutch school system, and to further clarify this point, we have adapted our initial sentence in Line 119 (now line 149-152).

Methods line 149-152:
To prevent that children’s ideas were influenced by earlier implemented PA programs and/or initiatives that promote physical activity in school (e.g. Healthy Schools, Jump-In, PLAYgrounds, JOGG), only schools that did not structurally provide such additional PA programs, were eligible to participate in our study.

Focus group composition – can you please provide more information about how many students from each level of PA behaviour ended up participating?

We have added this information in section 3.1. “Participant Characteristics” in the results.

Results, line 244-246
11 children were considered relatively high-active by their parents, 16 children medium-active and 16 children relatively low-active. Data on activity-levels of the other 9 children were not provided.

Discussion

· Lines 417 – 419: Is this supported by the results of the current study? Please clarify / explain.

We apologise for not being clear. This sentence refers to the results as shown in the second paragraph (line 340-357) of our results section, theme 3.2.4 ‘The role of the teacher’. Children indicate that in their experience some teachers already (are willing to) provide some forms of unstructured/spontaneous additional PA, but other teachers do not. To further illustrate the influence of teacher’s affinity with PA in this regard, we have added an additional quote in the results:

Results, line 352-353
“Teacher X. is more into gardening and being physically active, while teacher Y. focuses more on teaching the subjects”.

Furthermore, we have added a sentence in the discussion to clarify the connection to results.

Discussion, line 476-480:
For example, children indicated that they noticed vast differences between teachers with respect to what PA they offer, or their willingness to implement extra PA. Therefore, teachers’ priorities and affinities with PA seem to have a strong influence on whether or not children engage in (additional) PA at school.

· Lines 435 – 442: How are the results of these reviews related to your findings? Do they support or contradict them? Or, are you using these findings here to demonstrate the importance of student voice? Please explain. You touch on motivation benefits in the following paragraph but make no connections with the rest of the findings of these reviews.

The goal of the first paragraph within the subsection 4.2 “Child participation” was indeed to demonstrate an overview of the known benefits and the importance of involving children and giving them a voice when it comes to school-based health promotion. Based on the suggestions of reviewers #1, #2 and #3 to strengthen the rationale for examining children’s perspectives in intervention development, we decided that it is better to provide this information in our introduction section. Therefore, we removed it from our discussion section. Based on an earlier suggestion of the reviewer, we now discuss our findings related to student voice in the light of existing motivational theories.

· Line 454: I suggest using ‘establishing a child-PA-committee’ (rather than installing).

We have changed ‘installing’ to ‘establishing’.

· Lines 471 – 482: Thinking of this paragraph, are you suggesting that despite the importance of giving children voice, they sometimes lack the necessary knowledge to know whether they are realistic?

No, it was not our intention to suggest this. In this study the children were prompted to share all ideas that they might have to increase school-based PA, without asking them to consider their feasibility. Nevertheless, our experience during the focus groups was that children did take feasibility into account in some cases, as shown by the barriers (and the solutions to these barriers) recognised by the children. See results section, paragraph ‘3.2.6 External Barriers and Facilitators of PA’. In addition, this became clear from children’s discussion regarding the amount of extra time that should be spent on recess/PE/outside activities (Line 295-299, paragraph ‘Frequency and duration of PA’).

Therefore, with our summation that some ideas of the children were less feasible than others we did not mean to say that children do not have the ability to come up with realistic plans, but rather that some ideas are more difficult to implement than others, because it touches upon time and financial constraints outside their control, or even national policy. We have clarified this point in the discussion as follows:

Discussion, line 548-562

Many of children’s suggestions centred around expanding time of the existing PA curriculum, such as increasing the duration and frequency of PE, recess and occasional activities (e.g. sports day, school outings). These findings confirm the results of earlier studies and reflect the need of children to be more physically active during school time [37, 73, 82]. However, if we look at the myriad of barriers that schools currently experience, these forms of additional PA are difficult to realise in daily school practice, due to time constraints and the financial and material resources that are needed, e.g. hiring PA professionals, acquiring new (technological) play equipment, adapting playground facilities [41, 73]. This was also recognised by children in our study, whom, for example, worried about too little time remaining to finish school work. In the Netherlands, schools are accountable for reaching academic targets set by the government [83], and receive a budget from the government that can be used at the school’s discretion to hire staff and for maintenance of the school building [52]. In this sense, schools have some flexibility in deciding to spend (part of these) resources on facilitating more PA. However, to realise substantial extensions of the current PA curriculum, considerable investments and shifts in priorities of the school board or even the political level (such as installing legal provisions prescribing a minimum amount of PE in schools) seem necessary [81].

· Line 524: This sentence needs to be revised – do you mean ‘trials should be conducted…’

Indeed, we meant ‘trials should be conducted’, we have revised this in the manuscript.

· Given the focus of this study on ‘sustainability’, I suggest that more emphasis is given on this aspect in the discussion.

We thank the reviewer for his/her comment. It is true that this aspect was only a small aspect of our discussion, as the children only mentioned a few factors that would contribute to them sustaining in PA activities in their regard. We have added a paragraph that focusses specifically on the themes that were related to the sustainability of PA.

4.2. Sustainable participation in PA

Children mentioned several factors that are, according to them, important to stay motivated to participate in PA. We will now highlight the major themes that emerged. Children stressed the importance of involving them and giving them voices when it comes to additional PA in school. Children particularly recognised motivational benefits of student participation in relation to durable PA interventions, as they indicated that involving them in choosing and inventing activities will motivate them to sustain in PA participation in the longer run. This finding can be explained using the self-determination theory which postulates that three basic psychological needs (i.e. competence, autonomy and relatedness) need to be fulfilled in order to have children intrinsically motivated to engage and sustain in a health behaviour such as PA [42, 76]. Within this theory, the need for autonomy is a particularly important factor to attain high levels of intrinsic motivation [42]. However, our study also revealed that children questioned whether the school would actually be open to implementing their ideas, which indicates that it is currently not common to involve children in choosing and designing school-based activities.

On the other hand, children also indicated a need for guidance in their PA choices as they suggested that teachers can help them in choosing or designing PA activities. This aligns with the finding that, to successfully implement educational innovations and school health promotion programs, co-creation with all stakeholders is recommended, leading to increased ownership of the intervention in both students and teachers [22, 77, 78]. Moreover, teacher support and encouragement appeared important to keep children participating in school-based PA over time.

Our results indicate that 10- to 13-year-old children are very capable of inventing solutions to perceived barriers (e.g. sharing equipment, fundraising to collect money, using timetables, catch up on lost PA time). Hence, it seems of considerable value to involve children actively and make ‘additional PA in school’ a shared project of teachers and children. One specific way in which to increase student participation in school-based PA could be establishing a child-PA-committee, as suggested by some boys in our study. Lastly, children considered variation not only important to make PA enjoyable (see 4.3), but considered it also a major factor in keeping them interested to participate in additional PA in the long run. In particular, they highlighted the importance of variation in terms of (types of) activities and PA equipment, which indicates that it is important to adapt and change school-based PA programmes regularly.

Reviewer 2 Report

Thank you for inviting me to review this manuscript. This study involved focus groups with primary children to get their perspectives on how to increase physical activity in the school setting. The paper was very well written. Please find below comments and suggestions for improvement.

Introduction: It would be helpful to provide further detail in the introduction about the vast literature on school-based physical activity interventions, expanding further on the point made about how interventions are developed with limited involvement of school staff and children. How many previous interventions have been co-produced with staff and students and are these interventions more or less successful than other school-based interventions? Why would we expect co-produced interventions to be more successful? Perhaps referring to intervention development guidance that suggests involvement of stakeholders would be helpful here, see https://www.ncbi.nlm.nih.gov/pmc/articles/PMC4853546/ and https://bmcpublichealth.biomedcentral.com/articles/10.1186/s12889-017-4695-8.

A review for school-based physical activity which might be helpful to consider is https://onlinelibrary.wiley.com/doi/full/10.1111/obr.12352.

Method: Please elaborate on how schools were deemed to not provide PA programs. How did the authors define a PA program and how was this assessed?

The perspectives of other stakeholders such as teachers and parents were not obtained in the focus groups and this is a limitation. As noted by the children, teachers are essential to implementing school based PA opportunities in many cases. Lack of involvement of other stakeholders should be acknowledged as a limitation, or possibly as a future research opportunity to develop an intervention.

Similarly, several schools were included in this study. This is both a strength and limitation as the contexts of the schools included will vary and the possibilities for increasing PA will vary between schools. This should also be acknowledged.

Table 2 is extremely helpful in outlining the analysis approach used. One part of the table notes that 'V.B. checked and supplemented relevant text fragments after 'one or two transcripts'. Please specify after how many.

Please justify the use of thematic analysis and the theoretical underpinnings for using this approach

The titles of the themes could provide further detail on the content of the themes. For example, 'Additional PA in School according to children' could be 'Children's ideas and perspectives on incorporating additional PA in school', or similar. Several other theme titles could also be further specified, perhaps linking back to aims of the study more clearly.

Table 3 is very helpful and informative. It may also be helpful to provide a table outlining all of the themes and sub-themes for further clarity.

Parts of the discussion reads as a repetition of the results. This could be strengthened by reflecting more critically on how the findings add to previous research, outlining what this research adds in addition to similarities to other research.

In the conclusion it would be helpful to note that when designing PA opportunities in schools, the findings from the current study could be helpful to inform strategies but that further development of the strategies should include involvement from teachers and children (and other key stakeholders) to ensure relevance to local contexts.

Author Response

Thank you for inviting me to review this manuscript. This study involved focus groups with primary children to get their perspectives on how to increase physical activity in the school setting. The paper was very well written. Please find below comments and suggestions for improvement.

We thank the reviewer for his/her time and effort, and for the constructive feedback on our study. Below we list the comments and our responses to each comment.

Introduction: It would be helpful to provide further detail in the introduction about the vast literature on school-based physical activity interventions, expanding further on the point made about how interventions are developed with limited involvement of school staff and children. How many previous interventions have been co-produced with staff and students and are these interventions more or less successful than other school-based interventions? Why would we expect co-produced interventions to be more successful? Perhaps referring to intervention development guidance that suggests involvement of stakeholders would be helpful here, see

https://www.ncbi.nlm.nih.gov/pmc/articles/PMC4853546/ and https://bmcpublichealth.biomedcentral.com/articles/10.1186/s12889-017-4695-8.

We thank the reviewer for the suggestion to strengthen the rationale for co-created interventions, which was in line with the suggestion to expand on the rationale to investigate children’s perspectives made by reviewer#1. As suggested, in our introduction section we have elaborated more on the importance of including children’s perspectives and the potential benefits of co-created interventions, using the suggested papers of Wight et al (2015) and Hawkins (2017) (see highlighted text below). To the best of our knowledge, no previous studies compared the effectiveness of interventions that were co-produced by staff and students with those that were not. Therefore, we have included this as a recommendation for future research in our discussion section (line 603-606): Co-creation of PA promotion programs become more prevalent [78] and future effectiveness trials should be conducted to gain insight whether co-created PA promotion programs are more effective in improving physical activity and subsequently health and school-related outcomes.

Introduction, line 52-71

However, several systematic reviews concluded that school-based PA interventions often have only limited or short-term effects on children’s overall PA levels [13, 14]. A possible explanation for this finding is that the evaluated interventions are commonly developed with limited involvement of school staff and children and may therefore be insufficiently tailored to their specific circumstances, needs and wishes [15-17]. This could result in the development of inappropriate interventions resulting in sub-optimal implementation of and participation in these interventions, limiting their effectiveness [16, 18, 19]. In order to ensure that school-based PA interventions will succeed in daily school practice, they need to be relevant to local school contexts and appropriate for and attractive to the intended users (i.e. children and school staff) [15-17, 19-22]. A recent systematic review concluded that involving students in the development and implementation of school-based health activities, has positive effects on students’ motivation and attitude towards the activities, health-related outcomes, social interactions, as well as the school’s culture, climate, rules and policies [23]. Moreover, an earlier systematic review by Jacquez et al (2013) has shown that involving children and adolescents in research projects can lead to increased quality of research and higher chances of translating research outcomes into action in daily practice [17].

In the past years, several qualitative studies have investigated the perspectives of teachers and school administrators regarding primary school-based PA interventions [e.g. 24-27]. These studies provided valuable information on facilitators and barriers, mostly regarding the organisation and execution of PA. However, children are the ones who are expected to actually participate in the intervention, and it is very likely that their perspectives differ from the views of adult school staff.

Line 86-100
There is currently a lack of qualitative studies that focus specifically on PA opportunities in the primary school setting, and in particular involving the critical age group of 10-13 years, where, as mentioned above, PA tends to decline [7, 8]. Moreover, the qualitative studies investigating additional PA opportunities that have been conducted in the primary school setting often have a narrow focus, determined in advance by the researchers, such as PA promotion in recess [e.g. 35-37], or adapting the school physical environment [e.g. 38-40]. For example, a focus group study of Hyndman (2016) investigated what kind of school PA facilities 10-13-year-old children find appealing (e.g. a variety of recreational, sporting and adventure facilities). However, this study was limited to features of the physical environment and did not consider other ways of enhancing school-based PA. As previous studies focused on certain predefined aspects of the school day, it remains unclear how exactly PA opportunities can be increased within the broader primary school setting according to children. In particular, knowledge is lacking on what kind of activities would be appealing to children, as well as what they need to successfully engage in school-based PA. Therefore, a comprehensive investigation of children’s thoughts and ideas concerning PA promotion within the primary school setting in general is required.
Line 112-121
In sum, for the development of attractive and effective PA interventions in primary school, more knowledge of the perspectives, needs and wishes of children is needed. To the best of our knowledge, no qualitative studies have explicitly and comprehensively explored the multi-faceted ideas of 10-13-year-old children to increase PA in the primary school setting. Therefore, the aim of our study was to 1) explore 10-13-year-old primary schoolchildren’s perspectives on incorporating additional PA into the school day, in particular what practical ideas children have to be more physically active in school and what they think is important to successfully implement these ideas; and 2) to investigate children’s views on how to keep the proposed activities and ideas interesting and fun for a longer period of time. The results of this study can inform both research and practice regarding the development and implementation of future PA interventions aimed at 10-13-year-old children.

A review for school-based physical activity which might be helpful to consider is https://onlinelibrary.wiley.com/doi/full/10.1111/obr.12352.

Thank you for the relevant paper suggestion, we have used this reference in the revised version of our discussion.
Method: Please elaborate on how schools were deemed to not provide PA programs. How did the authors define a PA program and how was this assessed?

We thank the reviewer for raising this point. We apologise for being incomplete in our methods section. In the Netherlands, there are some national/municipal initiatives (e.g. Healthy Schools, Jump-In, PLAYgrounds, JOGG) that promote physical activity in schools, e.g. by providing PA interventions, school (playground) adaptations, or external coaches who organise PA activities during recess or after school hours. These initiatives are used in addition to or as part of the regular physical education curriculum and recess, and are based on previous research. The majority of schools in the Netherlands do not participate in these initiatives. We decided to exclude these schools, because it is likely that the ideas of children who participate in such PA interventions will be influenced by their PA experiences. We have now added information about these national/municipal initiatives in a new subsection of the methods, which elaborates on PA in the Dutch primary school system. Furthermore, we revised our explanation about the exclusion of schools for further clarification.

Methods (line 132-144):
2.2. PA in the Dutch primary school system

In the Netherlands, children aged 4 to 12 attend primary school (2 years of pre-school followed by grades 1 to 6). On average, children participate in an hour and a half of PE per week. There are no legal provisions issued by the government on the amount of PE that should be provided, but the guideline of the Ministry of Education, Culture and Science prescribes at least two to three lessons of PE per week. PE should either be taught by classroom teachers with a qualification for PE, or a PE specialist [51]. Schools are free choose their own recess hours, e.g. a short break in the morning and a longer one for lunch break, or two breaks of the same length [52]. Furthermore, schools receive a budget issued by the government that they can spend at their discretion on facilities or personnel, such as a PE teacher [53]. In addition to the regular PA curriculum, there are some national/municipal initiatives (e.g. Healthy Schools, Jump-In, PLAYgrounds, JOGG) that promote extra physical activity in schools, e.g. by providing PA interventions, school (playground) adaptations, or external coaches who organise PA activities during recess or after school hours.

Methods (line 149-152):
To prevent that children’s ideas were influenced by earlier implemented PA programs and/or initiatives that promote physical activity in school (e.g. Healthy Schools, Jump-In, PLAYgrounds, JOGG), only schools that did not structurally provide such additional PA programs, were eligible to participate in our study.

The perspectives of other stakeholders such as teachers and parents were not obtained in the focus groups and this is a limitation. As noted by the children, teachers are essential to implementing school based PA opportunities in many cases. Lack of involvement of other stakeholders should be acknowledged as a limitation, or possibly as a future research opportunity to develop an intervention.

The reviewer brings up a very valid point, for which we thank him/her. This study was actually part of a larger project which also investigated the perspectives of teachers and school principals on implementing more school-based PA (see Van den Berg et al., 2017: “It’s a Battle . . . You Want to Do It, but How Will You Get It Done?”: Teachers’ and Principals’ Perceptions of Implementing Additional Physical activity in School for Academic Performance). Considering the large amount of data collected, the researchers decided to present the perspectives of children and teachers/school staff in two separate papers. We have tried to connect the ideas of the children with the results of our and other previous studies on the perspectives of teachers and principals in our discussion section (see for example subsections XXX). We would like to emphasise that we agree with the reviewer that it is important to involve other stakeholders, and therefore, we have added a recommendation in our conclusion section.

Conclusion, line 643-647

Overall, our study provides a comprehensive overview of children’s voices regarding additional PA in school, which could be used to inform the development of future PA interventions aimed at increasing the activity levels of children in primary school. To ensure relevance to local contexts it is important that these strategies include the involvement of children, teachers and other key stakeholders.
Similarly, several schools were included in this study. This is both a strength and limitation as the contexts of the schools included will vary and the possibilities for increasing PA will vary between schools. This should also be acknowledged.

We thank the reviewer for raising this point. We have added this to our limitation section.

Limitations: Line 618-620.

First, due to the nature of qualitative research, our findings may have limited generalisability, for example to other school systems and to children with other ethnic backgrounds (most children were of Dutch origin). Moreover, we have to keep in mind that each school has its own context which influences children’s ideas regarding PA opportunities in their particular school.

Table 2 is extremely helpful in outlining the analysis approach used. One part of the table notes that 'V.B. checked and supplemented relevant text fragments after 'one or two transcripts'. Please specify after how many.

We apologise for this sentence being unclear. We meant to say that after researcher E.V. had either coded one or two manuscripts, the coding was checked by researcher V.B., depending on how soon V.B. was able to check the files. We have clarified the sentence as follows:

Table 2, line 234.
E.V. coded the remaining transcripts. Each time E.V. had coded either one or two transcripts, V.B. checked the manuscript(s) and supplemented relevant text fragments and assigned codes.

Please justify the use of thematic analysis and the theoretical underpinnings for using this approach.

We thank the reviewer for this suggestion and apologise for omitting a justification for the use thematic analysis in the methods. Thematic analysis as proposed by Braun & Clarke (2006) is a widely-used method for qualitative analysis, which is used for identifying, analysing and reporting patterns (themes) within data. To make sure children were unhindered by researchers’ previous notions about what might or might not be important in PA interventions, in our study we have chosen a ‘bottom-up’ inductive approach without an a-priori framework to guide and/or influence children’s answers. As thematic analysis is not theoretically bounded, it is a very suitable choice of method when using an inductive approach. We have further clarified the theoretical underpinnings of thematic analysis in the method section:

Methods, line 219-226
For the data analysis we followed the six steps of inductive thematic analysis of Braun and Clarke (2006) [59]. Thematic analysis as proposed by Braun & Clarke is a widely-used method for qualitative analysis, which is used for systematically identifying, analysing and reporting patterns (themes) within data (see audit trail in table 2). To make sure children were unhindered by researchers’ previous notions about what might or might not be important in PA interventions, in our study we chose a ‘bottom-up’ inductive approach without an a-priori framework to guide and/or influence children’s answers [60]. As thematic analysis is not theoretically bounded, it is a suitable choice of method when using an inductive approach.

The titles of the themes could provide further detail on the content of the themes. For example, 'Additional PA in School according to children' could be 'Children's ideas and perspectives on incorporating additional PA in school', or similar. Several other theme titles could also be further specified, perhaps linking back to aims of the study more clearly.

We thank the reviewer for this valuable suggestion. Based on the reviewer’s suggestion, we have adjusted most of the theme titles and one of the subthemes’ title in the results section:

Results
Theme 3.2.2. Children's Ideas and Perspectives on Incorporating Additional PA in School
Theme 3.2.3. Giving Children a Voice in Additional PA Participation

Theme 3.2.4. The Role of the Teacher in Providing Additional PA

Theme 3.2.5. Taking into Account the Differences between Children Related to PA Participation

Subtheme 3.2.5. Overcoming the differences between children in relation to PA participation

Theme 3.2.6. External Barriers and Facilitators of PA according to Children

Table 3 is very helpful and informative. It may also be helpful to provide a table outlining all of the themes and sub-themes for further clarity.

We thank the reviewer for this useful suggestion, we have now added an outline of all themes and subthemes in Figure 1 in the results section.

Figure 1. An overview of themes and subthemes resulting from the data analysis

Parts of the discussion reads as a repetition of the results. This could be strengthened by reflecting more critically on how the findings add to previous research, outlining what this research adds in addition to similarities to other research.

We thank the reviewer for his/her comment, which is also in line with the suggestion of reviewer #3 to put more emphasis on the novelty of our findings. We now have more clearly presented the findings that are novel in the discussion and have adjusted the statement about the focus of the discussion accordingly. We have also added a paragraph about similarities and contrast of general PA versus school-based PA, and discuss the importance of our Table 3 which provides a comprehensive overview of children’s voices with regard to preferred school-based PA. This has not been done before, and provides important information for future research. The revised sections now read as follows:

Discussion, line 436-443

To the best of our knowledge, this is the first study that comprehensively explored the views and needs of 10-13-year-old children to increase PA opportunities in the broader primary school setting. Children provided various ideas for additional PA and discussed important factors that need to be taken into account when developing feasible, enjoyable and sustainable school-based PA programs. In line with the socio-ecological model [63], children in our study mentioned factors related to opportunities to be physically active in school on multiple levels (i.e. individual level, social level, physical environment and school policy level). We will discuss the main findings of our study with a focus on their novelty and their implications for research and school practice.

Line 450-456

A novel finding of our study, expanding on the existing literature, was the importance of the perceived cognitive benefits of PA, specifically in the school setting. Children indicated that PA helps them to increase their motivation and focus, which was according to them particularly important given the long and uninterrupted bouts of sitting and/or working on school tasks during a school day. Children’s expressed motivations for and needs to be physically active during school time, reflect the importance and relevance of increasing PA opportunities in primary schools.

Line 526-546

4.3. General PA Promotion vs School-based PA

In line with earlier research on PA promotion in general (i.e. not specifically in the school setting), children in our study reported several factors that are important to make school-based PA enjoyable, such as variation [27, 37, 43, 66, 79], choice in PA activities [28, 33, 37, 42, 66], providing encouragement [73, 79] and new activities [27, 33], supervision of teachers [43, 80], and using technology [27, 37]. In contrast to earlier studies [72], children in our study did not emphasise the importance of competitive elements in PA, which might be due to differences in age of the participants (preadolescents in our study versus adolescents in [72]) or the setting of PA. Also, being physically active with friends has previously been reported by children as an important factor to make PA fun [27, 33, 79], but was not emphasised by the children in our study. This contrast between our and previous studies might be explained by the different settings of PA promotion, i.e. in school versus PA in general. Similar to earlier studies, children’s perceived barriers towards school-based PA included bad weather [36, 79] and lack of time to fully utilise PE [28, 29]. Furthermore, in line with teachers and principals [23, 81], the current school policies, lack of space and resources and financial constraints were also recognised by the children as barriers towards school-based PA. Overall, we can conclude that most facilitating factors and barriers in general PA promotion, also apply to PA promotion specifically in the school setting. Expanding on the existing literature on general PA promotion, our study provides a comprehensive and concrete overview of children’s voices with regard to preferred and enjoyable PA opportunities in school (see Table 3). This overview is of great value for future research and practice, as it can be used as a basis for developing school-based PA programmes that are appropriate for and attractive to 10-to-13-years old children.

Line 597-602

Although children emphasised the importance of taking into account individual preferences, another new finding of this study is that they also indicated that it is fine to perform PA activities that they consider less fun, as long as everyone’s preferences are represented once in a while. As such, it seems important for future research to invest more in active engagement/participation of children when developing PA programs, preferably together with the specific target group of the PA program [16, 77, 78].

Line 643-646
In the conclusion it would be helpful to note that when designing PA opportunities in schools, the findings from the current study could be helpful to inform strategies but that further development of the strategies should include involvement from teachers and children (and other key stakeholders) to ensure relevance to local contexts.

We thank the reviewer for this helpful addition, we have revised the recommendation in the conclusions accordingly.

Conclusions, line 644-647

Overall, our study provides a comprehensive overview of children’s voices regarding additional PA in school, which could be used to inform the development of future PA interventions aimed at increasing the activity levels of children in primary school

Reviewer 3 Report

ally

I think the authors of the paper have done a good job of writing up a qualitative paper. I think there is lots of evidence that illustrates rigour and good academic practice.

My main concern is content of results vs what has already gone ahead and just checking recent developments to be added in and consider against the results. Perhaps discriminating content more towards what is not known.

Introduction

I would like a critical synthesis of past work that goes beyond a description of study by study to identify a rationale for the study. Please just do a review of current work from 2017-2018 for any other studies and make sure a clear rationale is provided.

Past studies check if considered – A quick search reveal some articles not reference that I though may need consideration?

Barriers article

https://bmcpublichealth.biomedcentral.com/articles/10.1186/1471-2458-14-639

2018 qualitative article

https://www.tandfonline.com/doi/abs/10.1080/13573322.2016.1229290

2015 mixed methods

https://onlinelibrary.wiley.com/doi/full/10.1111/obr.12352

2017 qualitative review

https://onlinelibrary.wiley.com/doi/full/10.1111/obr.12562

2018 cross sectional work

https://journals.sagepub.com/doi/abs/10.1177/1059840518780300

Qualitative

2018

Laird

https://www.tandfonline.com/doi/abs/10.1080/17482631.2018.1435099

Ongoing work

http://www.publichealthwell.ie/node/308899?&content=resource&member=572160&catalogue=Campbell%20Reviews,Systematic%20Reviews&collection=none&tokens_complete=true

Other stake holder considerations e.g.,

https://www.tandfonline.com/doi/abs/10.1080/10888691.2016.1211482

Methods

Please add a paradigm and methodology for situating your work up front. Please cite a framework early on with this information like Obrien et al (2014) or Tong et al (2007) for focus group work.

In a supplementary file can we have examples of the steps and actual data so the reader can understand what is given in Table 2.

Results

Line 213-237 theme Positive motivation – fun, enough already, reasons for PA in the school, perceived benefits - has this not been covered before?

Theme 3.2.2 seems more like the rationale and need for work

Theme 3.2.3 and 3.2.4 – seem ok

Check 3.2.5 for previous content

3.2.6 external barriers – seems a little weak/underdeveloped compared to other themes.

Discussion

given the above - on resubmission I can consider

Author Response

I think the authors of the paper have done a good job of writing up a qualitative paper. I think there is lots of evidence that illustrates rigour and good academic practice.

My main concern is content of results vs what has already gone ahead and just checking recent developments to be added in and consider against the results. Perhaps discriminating content more towards what is not known.

We thank the reviewer for his/her positive assessment of our study and rigorous academic practice and the suggestion to discriminate content more towards what has not been previously found in research. Therefore, we have put more emphasis on the novel aspects of our study in the revised manuscript. Furthermore, considering the aim of our study, which was to gain comprehensive insight into all ideas that children have to be more physically active during the school day, we have chosen to report all aspects that children put forward as important in the focus group discussions in our results section, without discriminating content in advance.

Revised parts of the introduction (see below within our response to the reviewer’s questions about our introduction) and discussion:

Line 526-546

4.3. General PA Promotion vs School-based PA

In line with earlier research on PA promotion in general (i.e. not specifically in the school setting), children in our study reported several factors that are important to make school-based PA enjoyable, such as variation [27, 37, 43, 66, 79], choice in PA activities [28, 33, 37, 42, 66], providing encouragement [73, 79] and new activities [27, 33], supervision of teachers [43, 80], and using technology [27, 37]. In contrast to earlier studies [72], children in our study did not emphasise the importance of competitive elements in PA, which might be due to differences in age of the participants (preadolescents in our study versus adolescents in [72] or the setting of PA. Also, being physically active with friends has previously been reported by children as an important factor to make PA fun [27, 33, 79], but was not emphasised by the children in our study. This contrast between our and previous studies might be explained by the different settings of PA promotion, i.e. in school versus PA in general. Similar to earlier studies, children’s perceived barriers towards school-based PA included bad weather [36, 79] and lack of time to fully utilise PE [28, 29]. Furthermore, in line with teachers and principals [23, 81], the current school policies, lack of space and resources and financial constraints were also recognised by the children as barriers towards school-based PA. Overall, we can conclude that most facilitating factors and barriers in general PA promotion, also apply to PA promotion specifically in the school setting. Expanding on the existing literature on general PA promotion, our study provides a comprehensive and concrete overview of children’s voices with regard to preferred and enjoyable PA opportunities in school (see Table 3). This overview is of great value for future research and practice, as it can be used as a basis for developing school-based PA programmes that are appropriate for and attractive to 10-to-13-years old children.

Line 597-602

Although children emphasised the importance of taking into account individual preferences, another new finding of this study is that they also indicated that it is fine to perform PA activities that they consider less fun, as long as everyone’s preferences are represented once in a while. As such, it seems important for future research to invest more in active engagement/participation of children when developing PA programs, preferably together with the specific target group of the PA program [16, 77, 78].

Introduction

I would like a critical synthesis of past work that goes beyond a description of study by study to identify a rationale for the study. Please just do a review of current work from 2017-2018 for any other studies and make sure a clear rationale is provided.

We thank the reviewer for his/her suggestions to improve the introduction. The purpose of the study by study descriptions was to summarise what is already known from previous research about PA intervention preferences of children, in order to show that there is fragmented knowledge on children’s PA likes and dislikes within the primary school setting, and that no previous qualitative work focuses on the primary school setting in general. We now have provided extra clarification of this point in the revised manuscript and have added additional literature to provide a better overview of past work.

Introduction, line 52-71

However, several systematic reviews concluded that school-based PA interventions often have only limited or short-term effects on children’s overall PA levels [13, 14]. A possible explanation for this finding is that the evaluated interventions are commonly developed with limited involvement of school staff and children and may therefore be insufficiently tailored to their specific circumstances, needs and wishes [15-17]. This could result in the development of inappropriate interventions resulting in sub-optimal implementation of and participation in these interventions, limiting their effectiveness [16, 18, 19]. In order to ensure that school-based PA interventions will succeed in daily school practice, they need to be relevant to local school contexts and appropriate for and attractive to the intended users (i.e. children and school staff) [15-17, 19-22]. A recent systematic review concluded that involving students in the development and implementation of school-based health activities, has positive effects on students’ motivation and attitude towards the activities, health-related outcomes, social interactions, as well as the school’s culture, climate, rules and policies [23]. Moreover, an earlier systematic review by Jacquez et al (2013) has shown that involving children and adolescents in research projects can lead to increased quality of research and higher chances of translating research outcomes into action in daily practice [17].

In the past years, several qualitative studies have investigated the perspectives of teachers and school administrators regarding primary school-based PA interventions [e.g. 24-27]. These studies provided valuable information on facilitators and barriers, mostly regarding the organisation and execution of PA. However, children are the ones who are expected to actually participate in the intervention, and it is very likely that their perspectives differ from the views of adult school staff.

Line 86-100
There is currently a lack of qualitative studies that focus specifically on PA opportunities in the primary school setting, and in particular involving the critical age group of 10-13 years, where, as mentioned above, PA tends to decline [7, 8]. Moreover, the qualitative studies investigating additional PA opportunities that have been conducted in the primary school setting often have a narrow focus, determined in advance by the researchers, such as PA promotion in recess [e.g. 35-37], or adapting the school physical environment [e.g. 38-40]. For example, a focus group study of Hyndman (2016) investigated what kind of school PA facilities 10-13-year-old children find appealing (e.g. a variety of recreational, sporting and adventure facilities). However, this study was limited to features of the physical environment and did not consider other ways of enhancing school-based PA. As previous studies focused on certain predefined aspects of the school day, it remains unclear how exactly PA opportunities can be increased within the broader primary school setting according to children. In particular, knowledge is lacking on what kind of activities would be appealing to children, as well as what they need to successfully engage in school-based PA. Therefore, a comprehensive investigation of children’s thoughts and ideas concerning PA promotion within the primary school setting in general is required.

Past studies check if considered – A quick search reveal some articles not reference that I though may need consideration?

· Barriers article
https://bmcpublichealth.biomedcentral.com/articles/10.1186/1471-2458-14-639

·2018 qualitative article https://www.tandfonline.com/doi/abs/10.1080/13573322.2016.1229290

· 2015 mixed methods
https://onlinelibrary.wiley.com/doi/full/10.1111/obr.12352

· 2017 qualitative review
https://onlinelibrary.wiley.com/doi/full/10.1111/obr.12562

· 2018 cross sectional work
https://journals.sagepub.com/doi/abs/10.1177/1059840518780300

· 2018 Qualitative Laird
https://www.tandfonline.com/doi/abs/10.1080/17482631.2018.1435099

· Ongoing work
http://www.publichealthwell.ie/node/308899?&content=resource&member=572160&catalogue=Campbell%20Reviews,Systematic%20Reviews&collection=none&tokens_complete=true

· Other stake holder considerations e.g.,
https://www.tandfonline.com/doi/abs/10.1080/10888691.2016.1211482

We thank the reviewer for his/her paper suggestions. We agree that recently there has been much valuable research published on school physical activity interventions and the school physical activity environment. The body of literature spans many different age groups (varying from 4 to 18 years-old) and many different settings (i.e. high school, middle school, primary school or kindergarten; or even sub settings such as school playgrounds). In our study we have chosen to focus on what is known about the primary school setting and the PA preferences of age group of 10-13-year-old children. To make the focus of our study more clear to our readers, we have put more emphasis on the age group and primary school setting. Also, in line with the suggestions of the other reviewers, we have elaborated on the rationale of our study to investigate children’s perspectives in the introduction, also referring to papers that the reviewer has suggested.

 Methods

Please add a paradigm and methodology for situating your work up front. Please cite a framework early on with this information like Obrien et al (2014) or Tong et al (2007) for focus group work.

We thank the reviewer for this comment. We have indeed used the COREQ checklist by Tong et al (2007) while conducting and reporting our study but have overlooked to mention this in the method section. We have now added the following sentence:
Methods, line 219-221

For the data analysis we followed the six steps of inductive thematic analysis of Braun and Clarke (2006) [59], and we used the checklist for qualitative reporting of interviews and focus groups as proposed by Tong et al [60] to report the data.
As we also mentioned in our response above, in order to stay as close to the ideas and voices of children as possible, we have deliberately chosen to use a bottom-up approach and employ an inductive methodology using thematic analysis and open coding, meaning we did not analyse the data using a pre-existing theory/framework. We have now elaborated a bit more on this choice in the methods section. Furthermore, we added a justification for the use of thematic analysis

Methods, line 221-227

Thematic analysis as proposed by Braun & Clarke is a widely-used method for qualitative analysis, which is used for systematically identifying, analysing and reporting patterns (themes) within data (see audit trail in table 2). To make sure children were unhindered by researchers’ previous notions about what might or might not be important in PA interventions, in our study we chose a ‘bottom-up’ inductive approach without an a-priori framework to guide and/or influence children’s answers [61]. As thematic analysis is not theoretically bounded, it is a suitable choice of method when using an inductive approach.

In a supplementary file can we have examples of the steps and actual data so the reader can understand what is given in Table 2.

We apologise if some of the steps in Table 2 may have been unclear and thank the reviewer for his/her suggestion to add examples of the steps. To make Table 2 more informative for readers we have added an extra column with examples of each step when applicable.

Table 2. Audit trail of the data analysis, following the steps of Braun and Clarke [59].

Step

Description

Examples

1. Familiarising   yourself with the data

V.B. and   research assistant transcribed focus group data verbatim.

E.V. and V.B.   read all transcripts and discussed initial ideas and interesting features.

Children seemed   to have many ideas regarding PE, active games, recess activities and   activities outside the school building

2. Generating   initial codes

E.V. and V.B.   open coded two transcripts independently, selecting relevant text fragments   and ascribing   initial codes. After each transcript, E.V.   and V.B. compared their work and   discussed until consensus was reached.

E.V. coded the remaining   transcripts. Each time E.V. had coded either one or two transcripts, V.B.   checked the manuscript(s) and supplemented relevant text fragments and   assigned codes. Discrepancies were   discussed until consensus was reached. This   process was repeated until E.V. and V.B. concluded that coding the transcript   yielded no significant new codes in relation to the previously coded   transcripts. This was the case after coding four focus groups with boys and   five focus groups with girls.

Initial code   examples: “PA barrier: weather”, “PA   facilitator: teacher”, “Resources: playing   equipment”, ‘”PA motivation: health”

The coders   discussed whether the text fragment “I would like to play more sports in   school” should be coded as “Need to be more active”, or “Need for more PE”

The coders   discussed whether children meant the same thing when they indicated that they   prefer more ‘workshops’ or ‘clinics’

3. Searching for   themes

E.V. re-read all   coded data, comparing the coded extracts to the assigned code names. Similar   codes   were grouped together into initial (sub)categories. The collated text   fragments of (sub)categories were read and re-read to identify potential   overarching themes.

The codes “doing   the same activities” and “alternate location of PA” were grouped together in   the subtheme “variation”, and theme “characteristics of additional PA”.

4. Reviewing   themes

E.V. formulated   a preliminary map of the main (sub)themes and how they related to each other.     The coherence and distinctness of the themes   and subcategories were first discussed and revised together with V.B., and   subsequently within the larger research team (E.V., V.B., A.S. and M.C.).

Initial themes   that were identified: “Children’s motivations”, “Characteristics of   additional PA”, “Influences on enjoyable   PA”   (Choice, Personal preferences, Inclusion, Supervision), “External   barriers and facilitators”

5. Defining and   naming themes

Going back and   forth   between all data, E.V. refined the content   of each (sub)theme, collated significant quotes and wrote a first draft of   the results.

E.V. and V.B. reflected   on the first draft of the results in detail until consensus about   clear   definitions of the (sub)themes was reached.

The theme   “characteristics of additional PA” was   revised into “Additional PA according to   children”, with subthemes ‘variation’, ‘location’ etc

Some of the   subthemes were revised into a main theme, for example subtheme “Inclusion”   was revised into  main theme: “Taking   into account the   differences between children”, with   subthemes ‘Perceived differences’ and   ‘Strategies to include everyone’

6. Producing the   report

E.V. refined and   completed the report of the data analysis with input from V.B., A.S., M.C. and   R.G.

Note: E.V.: Eline Vos, V.B.: Vera van den Berg, A.S.: Amika Singh, M.C.: Mai Chin A Paw, R.G.: Renate de Groot.

Results

Line 213-237 theme Positive motivation – fun, enough already, reasons for PA in the school, perceived benefits - has this not been covered before?

The reviewer is right that some of these motivations for engaging in PA have been found in previous studies, and we elaborate on this in the discussion. There, we also explain how our findings on children’s perceived cognitive benefits of PA (specifically in the school setting) expand on existing literature. The section within the discussion is as follows:

Line 450-456

A novel finding of our study, expanding on the existing literature, was the importance of the perceived cognitive benefits of PA, specifically in the school setting. Children indicated that PA helps them to increase their motivation and focus, which was according to them particularly important given the long and uninterrupted bouts of sitting and/or working on school tasks during a school day. Children’s expressed motivations for and needs to be physically active during school time, reflect the importance and relevance of increasing PA opportunities in primary schools.

Theme 3.2.2 seems more like the rationale and need for work

We apologize but do not understand what the reviewer means to say?

Theme 3.2.3 and 3.2.4 – seem ok

We thank the reviewer for his/her approval of these sections in the results.

Check 3.2.5 for previous content

Apologies but we are not sure what the reviewer means to say. However, we have rephrased the title of this theme and now put extra emphasis on the strategies that were proposed by children to ensure that everyone is included in PA, as these findings have not yet been reported elsewhere. The section in the discussion is now as follows:

Line 586-606

4.5. Tailored PA Programs

Also worthwhile reporting is the apparent contradiction between children’s need for ‘tailored PA’ and ‘including everyone in PA’. On the one hand children indicated the importance of taking into account their individual needs and preferences, while on the other hand they emphasised that efforts must be made to make sure that all children can participate in additional PA in school. Related to the first, earlier studies have repeatedly recommended to develop individually tailored PA programs [15, 27, 41, 43, 65]. However, these recommendations are rarely followed-up since the development of such PA programs is very difficult due to considerable heterogeneity in school populations and corresponding logistical challenges [41, 65]. Hence, most PA programs are still of a ‘one size fits all’ type. Our results indicate that it is important to find a middle way in this paradox i.e. finding solutions to meet individual preferences within a group setting.

Although children emphasised the importance of taking into account individual preferences, another new finding of this study is that they also indicated that it is fine to perform PA activities that they consider less fun, as long as everyone’s preferences are represented once in a while. As such, it seems important for future research to invest more in active engagement/participation of children when developing PA programs, preferably together with the specific target group of the PA program [16, 77, 78]. It is important that children can discuss the content of the PA program and find compromises when preferences differ. Co-creation of PA promotion programs become more prevalent [78] and future effectiveness trials should be conducted to gain insight whether co-created PA promotion programs are more effective in improving physical activity and subsequently health and school-related outcomes.

3.2.6 external barriers – seems a little weak/underdeveloped compared to other themes.
We thank the reviewer for this comment. This can partly be attributed to the fact that children were not explicitly asked after barriers and facilitators of school-based PA in the focus group discussions. Nonetheless, some important barriers according to children came up during the discussion. Given the fact that they were very relevant for the implementation of the children’s PA ideas (see Table 3) we decided to report them. Furthermore, we found it interesting that children came up with unprompted solutions for these perceived barriers, which can also be used to inform future PA interventions.

Discussion

given the above - on resubmission I can consider

We thank the reviewer in advance for his/her reconsideration.

Round 2

Reviewer 3 Report

Thank you for responding positively to my past comments. Best wishes of your future research